# Mixture of Concept Bottleneck Experts

**Francesco De Santis** [1]   **Gabriele Ciravegna** [2]   **Giovanni De Felice** [3]   **Arianna Casanova** [4]   **Francesco Giannini** [5]
**Michelangelo Diligenti** [6]   **Johannes Schneider** [4]   **Danilo Giordano** [1]   **Mateo Espinosa Zarlenga** [7]   **Pietro Barbiero** [8]

## Abstract

Concept Bottleneck Models (CBMs) promote interpretability by grounding predictions in human-understandable concepts. However, existing CBMs typically constrain their task predictor to a single expression whose functional form is set a priori, limiting both predictive accuracy and adaptability to diverse user needs. We propose Mixture of Concept Bottleneck Experts (M-CBEs), a framework that generalizes existing CBMs along two dimensions: the number of expressions, referred to as experts, employed by the task predictor to map concepts to the task, and the functional form each expression takes, thus exposing an underexplored region of this design space. We investigate this region by instantiating two novel models: Linear M-CBE, which learns a finite set of linear expressions, and Symbolic M-CBE, which leverages symbolic regression to discover expert functions from data subject to user-specified operator vocabularies. Empirical evaluation demonstrates that varying the number of expressions and their functional form provides a robust framework for navigating the accuracy-interpretability trade-off.

## 1. Introduction

In recent years, Deep Learning (DL) models have achieved remarkable performance across a wide range of tasks, yet their growing complexity and opacity (Rudin, 2019; Colombini et al., 2025) prevent their adoption in high-stakes domains where transparency is essential (EUGDPR, 2017; Durán & Jongsma, 2021; Act, 2024). Concept Bottleneck Models (CBMs) (Koh et al., 2020) have emerged as a promising approach to align models with human reasoning.

[1]Polytechnic of Torino [2]Intesa Sanpaolo Innovation Center [3]USI [4]University of Liechtenstein [5]University of Pisa [6]University of Siena [7]University of Oxford [8]IBM Research. Correspondence to: Francesco De Santis <francesco.desantis@polito.it>.

*Proceedings of the $43^{rd}$ International Conference on Machine Learning*, Seoul, South Korea. PMLR 306, 2026. Copyright 2026 by the author(s).

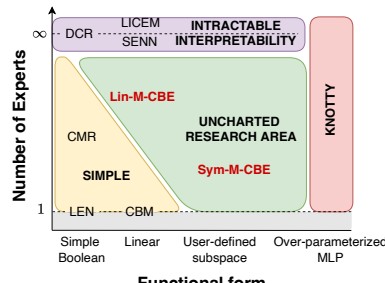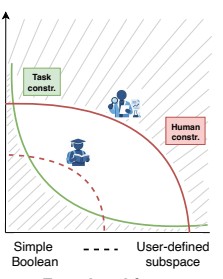

*Figure 1.* **Left**: The plane defined by functional form and number of experts. Different CBMs occupy distinct positions in this space, with regions highlighted in different colors. We indicate our newly proposed instantiations in red. **Right**: The red curve denotes human constraints, and the green curve denotes task constraints. The region bounded by the two curves represents the set of feasible models.

CBMs decompose prediction into two stages: a *concept encoder* that maps raw inputs to human-interpretable variables, called *concepts* (e.g., "*round*", "*red*"), and a *task predictor* that maps these concepts into the task variable.

While substantial effort has been devoted to defining and learning meaningful concepts (Oikarinen et al., 2023; Dominici et al., 2025; De Felice et al., 2025), little attention has been paid to the design of the task predictor. Indeed, most CBMs constrain the task predictor to provide predictions through a unique, global function (Koh et al., 2020; Vandenhirtz et al., 2024; Ciravegna et al., 2023; Marconato et al., 2022), restricting expressiveness and often failing to capture the true concept-to-task generating process (Mahinpei et al., 2021; Espinosa Zarlenga et al., 2022). A promising way to improve the expressiveness of task predictors is to use multiple specialized functions, as in Mixture of Experts (MoE) models (Jacobs et al., 1991; Shazeer et al., 2017).

Traditionally, MoE models have been designed to route inputs to experts so as to maximize predictive performance under computational constraints (Lepikhin et al., 2020; Artetxe et al., 2021). However, in an interpretability-oriented framework, this objective naturally translates into jointly optimizing predictive performance and interpretability. However, transposing MoEs to this new setting requires more than just expert routing; it demands explicit control over the *functional form* of the experts, the class of expressions map-

ping concepts to targets (e.g., Boolean or linear). Since interpretability is inherently user-dependent (Lipton, 2018; Miller, 2019; Gkintoni et al., 2025), fixing the functional form *a priori* constrains interpretability and may unnecessarily sacrifice predictive performance, as we show in Section 5. Consequently, enabling flexible control over both the functional form and the number of experts is central for achieving the optimal accuracy-interpretability trade-off.

**Contributions.** We introduce a new framework, Mixture of Concept Bottleneck Experts (M-CBEs), which generalizes CBMs by modeling the task predictor as a mixture of specialized functions (experts) with controllable functional forms. We adopt expression trees as our modeling abstraction, for a rigorous yet flexible definition of each function. We show that several existing concept-based models are special cases of our framework, while a substantial area remains unexplored (Figure 1, Left). To address this, we propose the Linear M-CBE model, which generalizes the linear CBM to multiple experts. Beyond instantiating specific functions, M-CBEs can be specified with the set of operators the user understands (e.g., $+, \times, \exp$). We demonstrate this capability with the Symbolic M-CBE model, which leverages symbolic regression to automatically discover the optimal expert functions using user-defined operators.

We empirically validate M-CBEs on classification and regression benchmarks, showing: i) navigating the design space allows finding the intersection of human and task constraints (Figure 1, Right), matching black-box accuracy without compromising interpretability; ii) algebraic forms ensure scalability outperforming rigid Boolean logic in high-dimensional concept spaces (up to + 65%); iii) multiple experts compensate for incomplete concept bottlenecks or varying task logic; iv) finite discrete expressions ensure global interpretability allowing user inspection; and v) while M-CBEs adapts to any functional form requested, Symbolic M-CBE also supports post-hoc adaptation, allowing users to modify operator vocabularies without retraining. The code to reproduce all experiments is available at `Mixture-of-CB-Experts`.

## 2. Preliminaries

**Concept Bottleneck Models.** Let $X$ and $Y$ be random variables representing the input and task, taking values in spaces $\mathcal{X}$ and $\mathcal{Y}$, respectively. We denote realizations by $x \in \mathcal{X}$ and $y \in \mathcal{Y}$. CBMs (Koh et al., 2020) introduce intermediate and human-interpretable concepts $C$ mapping into $\mathcal{C} \subseteq \mathbb{R}^k$, that mediates the relationship between $X$ and $Y$. The model is trained to approximate the joint distribution:

$$p(y, c \mid x) = p(y \mid c)\, p(c \mid x), \qquad (1)$$

where $c \in \mathcal{C}$, $p(c \mid x)$ is the *concept encoder* that predicts concepts from raw inputs, and $p(y \mid c)$ is the *task predictor*

that maps concepts to the task variable. This allows us to obtain a task predictor whose input is *semantically transparent*, as it operates solely on the interpretable concepts.

To increase the predictive accuracy of the task predictor, some concept-based models (e.g., Espinosa Zarlenga et al. (2022); Ismail et al. (2023)) leak extra information from $x$, achieving higher accuracy at the expense of semantic transparency.

**Symbolic Regression.** Given a dataset $\{(x^{(i)}, y^{(i)})\}_{i=1}^{N}$, Symbolic Regression (SR) searches the space of mathematical expressions to find the function $f : \mathcal{X} \to \mathcal{Y}$ closest to the true data-generating process (Schmidt & Lipson, 2009). SR methods employ heuristic search strategies, e.g. genetic algorithms, to evolve populations of expressions within a multi-objective optimization framework that jointly minimizes prediction error and expression complexity (Langley, 1979; Langley et al., 1981; Koza, 1994; Cranmer, 2023).

## 3. Mixture of Concept Bottleneck Experts

This section presents Mixture of Concept Bottleneck Experts (M-CBEs), a class of models that improves upon existing CBMs in two ways: (i) by allowing flexible, user-aligned functional forms, ensuring that the task predictor can only retrieve functions the user is capable of understanding while tailoring expressiveness to the problem at hand; and (ii) by operating over ensembles of expressions (experts), enabling the task predictor to match accuracy through multiple simpler expressions or to handle inputs requiring distinct reasoning patterns.

M-CBEs represents each task-predictor function as an expression tree and models it as a random variable $T$ (with realization $t$) conditioned on the input $x$. Conditioning $T$ on $X$ allows the model to dynamically select different expressions across inputs, maintaining high accuracy while preserving semantic transparency (Barbiero et al., 2023; De Santis et al., 2025a; De Felice et al., 2025). Indeed, the task prediction depends on the expression tree $t$, which operates solely on the predicted concepts $c$. This formulation induces the probabilistic graphical model (PGM) illustrated in Figure 2 (Left). While our framework shares structural similarities with the PGM proposed by Debot et al. (2024), we do not restrict $T$ to represent a Boolean expression; instead, we generalize the model to accommodate arbitrary functional forms. Accordingly, the joint distribution factorizes as:

$$p(x, c, t, y) = p(x)\, p(c \mid x)\, p(t \mid x)\, p(y \mid c, t) \qquad (2)$$

The first factor, $p(c \mid x)$, represents the *concept encoder*, while $p(t \mid x)$ defines the distribution over expression trees. The final term, $p(y \mid c, t)$, is the *task predictor*, which specifies the target distribution conditioned on both predicted concepts and the selected tree.

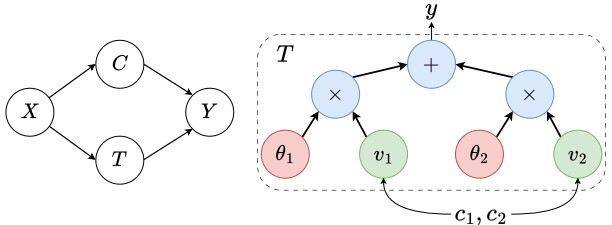

*Figure 2.* **Left:** PGM representing the assumed generative process for M-CBEs. **Right:** Example of an expression tree ($T$) representing a linear expression. Green nodes are placeholder variables $V$, blue nodes are operators $O$ drawn from the vocabulary $\mathcal{W}$, and red nodes are learnable parameters $\Theta$. At inference time, each placeholder $v_i$ is instantiated with the corresponding predicted concept $c_i$.

## 3.1. Expression Trees in M-CBEs.

An expression tree (Mitchell, 1991) is a directed acyclic graph (DAG) representing a mathematical expression (Figure 2, right). We define a class $\mathcal{T}$ of expression trees w.r.t. a vocabulary of operations $\mathcal{W}$ via admissible edge configurations $\mathcal{E}$ on a set of nodes $\mathcal{N} = \mathcal{V} \cup \mathcal{O} \cup \mathcal{P}$, with:
(i) $\mathcal{V}$: set of *input nodes* instantiated with specific **concepts**.
(ii) $\mathcal{O}$: set of *operator nodes* (e.g., $\{\times, \sin, \exp\}$) from $\mathcal{W}$.
(iii) $\mathcal{P}$: set of *parameter nodes* representing real numbers.

Let $O, E, \Theta,$ and $V$ denote the random variables for operators, edges, parameters, and placeholder inputs, taking values in $\mathcal{O}, \mathcal{E}, \mathcal{P},$ and $\mathcal{V}$, respectively. An expression tree is defined by the tuple $t = (o, e, \theta, v)$, where $o, e, \theta$ and $v$ represent specific realizations of these random variables. Notably, we decouple the input variable $V$ from the concepts $C$ predicted by an encoder, allowing the expression tree to selectively learn and operate on a subset of relevant concepts. We denote the space of functions representable by trees in class $\mathcal{T}$ as $\mathcal{H}(\mathcal{T})$. Given concept values $c$ assigned to the input variables, evaluating the tree yields a prediction $y = f_t(c)$, where $f_t$ represents the function defined by the expression tree $t$.

For instance, the set of linear functions can be represented by selecting $\mathcal{W} = \{+, \times\}$, where $+$ is the binary summation and $\times$ the binary multiplication, and having $O$ and $E$ only allowing a multiplication layer between pairs of input and parameter nodes, and a unique summation on top (cf. Figure 2). Each selection of $\mathcal{P}$ defines a different linear function. We refer to the set of expression trees whose represented functions are linear as $\mathcal{T}_{\text{lin}}$.

## 3.2. Design Choices in M-CBEs

**Functional form.** Navigation through the functional form dimension is realized by performing different inferences over the generative process. For instance, the user can fix $o$, $e$, and $v$, reducing the problem to a parametric linear

function where only the parameters $\theta$ are learned from data. Alternatively, one can fix only part of the structure by setting a sub-expression (specific $o$ and $e$ for a subset of nodes) while learning the remaining operators, edges and subset of concepts $v$ from data, enabling the incorporation of domain knowledge without fully specifying the expression tree. Finally, one can provide only the vocabulary $\mathcal{W}$ of interpretable operators while learning the entire expression tree $t$ from data, subject to the constraint that all operators belong to $\mathcal{W}$. Each scenario is naturally accommodated through appropriate conditioning and marginalization within our probabilistic framework.

**Number of Experts.** Models based on a finite set of prototypes are often considered a strict requirement to ensure global interpretability (Rudin, 2019; Rudin et al., 2022). Indeed, in the proposed framework we consider a finite set of expression trees, allowing the task predictor to be inspected and validated by the user in finite time — along with other important properties discussed in Appendix A. To realize this, we model $p(t \mid x)$ as a mixture obtained by marginalizing over $M$ discrete indices:

$$p(t \mid x) = \sum_{m=1}^{M} p(t \mid m)\, p(m \mid x), \qquad (3)$$

where $p(m \mid x)$, called the *selector*, is a learnable distribution that routes each input $x$ to a specific index $m$ (Debot et al., 2024). Given Eq. 3, multiple expression trees $t$ might be associated to the same index $m$, ultimately reducing the interpretability of the final prediction (Breiman, 2001; Meinshausen, 2010). Therefore, to ensure that each index $m$ corresponds to a unique expression tree, $p(t \mid m)$ is modeled as a degenerate distribution that places all its probability mass on the $m$-th expression tree, $\delta_t(t_m)$. Hence, given a realization $m$, there is exactly one expression tree $t_m$ associated with it. By varying $M$, we adjust the number of experts, balancing expressiveness (obtained with high values of $M$) and interpretability (obtained by lower values of $M$). Under the above assumptions Equation (2) can be rewritten in conditional form as:

$$p(y, c \mid x) = p(c \mid x) \sum_{m=1}^{M} p(m \mid x)\, p(y \mid c, t_m). \qquad (4)$$

**Expressiveness-Interpretability trade-off.** The flexibility on the functional form allows M-CBEs to connect with different variants of the universal approximation theorem (Cybenko, 1989). Moreover, $\mathcal{T}$ and the number of experts $M$ play a crucial role in calibrating the trade-off between the expressiveness and the complexity of the functions represented by any expert, as we demonstrate in the following.

**Proposition 1.** *(1) Let* $\mathcal{T}' = \{t \in \mathcal{T} : o \in \{+, \times, \sigma\}\}$, *being $\sigma$ a unary non-polynomial operation. Then for each*

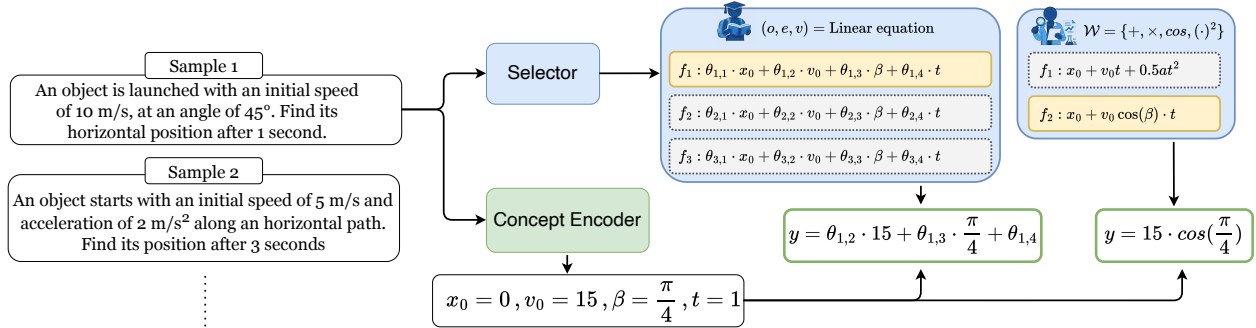

*Figure 3.* Overview of M-CBEs. The selector identifies the appropriate expression based on the input. Simultaneously, the Concept Encoder predicts the underlying physical concepts $(x_0, v_0, \alpha, t)$ from the raw input. The architecture accommodates different user mental models by allowing functional forms to be represented as either parametric linear equations or symbolic expressions restricted to interpretable operators. The final output $y$ is obtained by executing the selected function with the predicted concepts as arguments.

*continuous function $f$ and $\epsilon > 0$ there exists an M-CBE $f^*$ over $\mathcal{T}'$ with $M = 1$ experts, such that $\|f - f^*\|_\infty < \epsilon$. (2) Let $\mathcal{T}_{lin}$ denote the set of expression trees whose represented functions are linear. For each continuous function $f$ and $\epsilon > 0$, there exists a certain $M > 0$ and an M-CBE $f^*$ over $\mathcal{T}_{lin}$ with $M$ experts, such that $\|f - f^*\|_\infty < \epsilon$.*

See Appendix B for a proof.

Clearly, to approximate a function with only linear experts, $M$ cannot be fixed a priori, but any function may require a different one. Next, we highlight how the approximation error can be bound given a set of polynomial experts.

**Proposition 2.** *Let $\Omega \subseteq [a,b]^n$ be a convex finite domain, $f : \Omega \to \mathbb{R}$ a function of class $C^\infty$, and $\Omega$ be partitioned into $M$ non-overlapping subspaces with equal measure, being $I_m$ the indicator function of the $m$-th subspace. Given the space of expressions trees computing polynomial functions $\mathcal{T}_p$, there exists a selection of polynomials $f_t \in \mathcal{H}(\mathcal{T}_p), t = 1, \ldots, M$ with maximum degree $k$, such that their composition $f^\star(x) = \sum_{t=1}^{M} f_t(x) \cdot I_t(x)$ approximates $f$ with error:*

$$Error_{max} = \|f - f^\star\|_\infty \leq C \cdot \frac{(b-a)^{k+1}}{M^{\frac{k+1}{n}}} \cdot \|f^{(k+1)}\|,$$

*where $C$ is a fixed constant, $\|f^{(k+1)}\|$ is a Sobolev norm measuring how regular the target function is in terms of its $k+1$-th partial derivative.*

See Appendix C for a proof.

Interestingly, Theorem 2 provides an estimation of the approximation errors for the general case of polynomial experts. For instance, assuming the experts are linear ($k = 1$) the approximation error is bound by $C \cdot \frac{(b-a)^2}{M^{\frac{2}{n}}} \cdot \|f^{(2)}\|$. This means that the error is low if the target function $f$ is

close to linear, e.g., $\|f^{(2)}\| \approx 0$, and that it can be arbitrarily lowered by increasing the number of experts $M$. We also notice that Theorem 2 assumes experts are selected from equal-measure subspaces, which corresponds to having each expert assigned to similar portion of the data in the empirical distribution. Even if beyond the scope of this paper, this property could be enforced via an appropriate regularization mechanism in the selector.

## 4. Instantiations of M-CBE

M-CBEs encompass a broad spectrum of CBM architectures, including existing concept-based methods that can be interpreted as particular choices within our unified framework. For a mapping between specific choices in our framework and existing CBMs, we refer to Appendix D.

Beyond capturing existing models, M-CBEs enables the creation of CBMs that explore previously unconsidered configurations. These instances model $p(t \mid x)$ as a mixture of expression trees (Equation (3)), ensuring predictive accuracy while maintaining global interpretability. For instance, models employing linear functions with more than one expert have yet to be defined; at the same time, functional forms different from linear or Boolean expressions have yet to be considered. To fill this gap, we first propose a model fixing the *structure* $(o, e, v)$ of the expression tree to a parametric linear form over all concepts, while learning only its parameters. Second, we consider a scenario in which the user aims to discover the structure of an expression while retaining control by restricting the vocabulary to a set of known, admissible operators $\mathcal{W}$. Under this setting, the functional form is fully determined by the chosen operators $\mathcal{W}$. Figure 3 illustrates both instantiations.

**Linear M-CBE (Lin-M-CBE).** Our first instantiation extends the standard CBM (Koh et al., 2020) to accommodate a mixture of specialized linear expressions while retaining

its original functional form. Lin-M-CBE restricts the expression tree to a linear structure but permits selection from a finite set of $M$ distinct linear expressions. The conditional distribution in Equation (4) can be rewritten as:

$$p(y, c \mid x) = p(c \mid x) \sum_{m=1}^{M} p(m \mid x) \, p(y \mid c, o_l, e_l, v, \theta_m) \,, \quad (5)$$

where $o_l$ and $e_l$ are fixed realizations defining a linear structure, and $v$ instantiates the function over all concepts. For example, in a regression task $p(y \mid c, o_l, e_l, v, \theta_m) = \mathcal{N}\big(y; f_{(o_l, e_l, v)}(c, \theta_m), \sigma^2\big)$, where $\sigma^2$ represents the variance, $f_{(o_l, e_l, v)}(c; \theta_m) = w_m^\top c + b_m$ and $\theta_m = (w_m, b_m)$. The model selects among these $M$ expressions based on $p(m \mid x)$, allowing flexibility in how concepts are combined to make predictions while maintaining a linear structure.

**Symbolic M-CBE (Sym-M-CBE).** Beyond specifying the expression tree, a user can constrain the expression by only defining the vocabulary $\mathcal{W}$ of interpretable operators. The conditional distribution retains the structure of Equation (4). Learning the expression trees can be approached in two ways. The first uses differentiable symbolic regression methods (Martius & Lampert, 2016) to learn each $t_m$ end-to-end with the selector and concept encoder. The second approach distills symbolic expressions from placeholder differentiable black-boxes (Alaa & Van der Schaar, 2019; Cranmer et al., 2020; Liu et al., 2025). Specifically, the latter approach decouples learning into three stages: (i) jointly training the concept encoder, selector, and placeholder black-box predictors (e.g., MLPs), allowing the data to be partitioned into $M$ mechanism-specific subsets; (ii) applying symbolic regression to each subset to recover the corresponding expression tree $t_m$; and (iii) replacing the placeholders with the extracted symbolic expressions and fine-tuning the expression parameters $\theta_m$ end-to-end with the concept encoder and selector. This decoupled approach enables user adaptability: after initial training, different users can specify their own requirements to obtain expression trees aligned with their domain expertise, without retraining the encoder or selector. For this reason, we adopt this second strategy. To distill symbolic expressions from the placeholder black-box predictors, we use the multi-population evolutionary algorithm provided by PySR (Cranmer, 2023). The operator set $\mathcal{W}$ is entirely determined by the user; to maintain interpretability, users can eliminate operators deemed inappropriate for specific concepts (e.g., removing $\exp$ if an exponential relationship is implausible for the domain at hand). In our experiments, we use $\mathcal{W} = \{+, -, \times\}$ for classification tasks and $\mathcal{W} = \{+, -, \times, \sin, \cos, \exp, \log, \tan, \tanh, x^2, x^3, \sqrt{\cdot}, (\cdot)^{-1}\}$ for regression tasks. An ablation comparing with KANs (Liu et al., 2025) is provided in Appendix E.

### 4.1. Training Objective

Let $\mathcal{D} = \{(x^{(i)}, c^{(i)}, y^{(i)})\}_{i=1}^{N}$ be a concept-annotated dataset of size $N$. Following the factorization in Equation (5), we train Lin-M-CBE by maximizing the corresponding log-likelihood:

$$\mathcal{L} = \sum_{i=1}^{N} \Big[ \log p(c^{(i)} \mid x^{(i)}) +$$
$$+ \log \sum_{m=1}^{M} p(m \mid x^{(i)}) \, p(y^{(i)} \mid c^{(i)}, o, e, v, \theta_m) \Big],$$

where the first term corresponds to the concept prediction loss, and the second term corresponds to the task prediction loss. As previously noted, $o = o_l$ and $e = e_l$ specifies a linear structure. For Sym-M-CBE, the first training stage jointly learns the concept encoder, selector, and a set of placeholder predictors by optimizing the same objective, where the predictors are implemented as MLPs with different operator and edge configurations $(o, e)$. In the second stage, the training data assigned to each expression is used to recover a symbolic expression tree via symbolic regression. In the final stage, the learned symbolic expressions replace the placeholders, and their parameters, $\theta_m$, are fine-tuned end-to-end with the concept encoder and selector. Further training details are provided in Appendix F.

## 5. Experimental Results

We empirically validate the benefits of exploring the design plane. Our experiments assess four key properties of interpretable models: (i) **Accuracy vs. Interpretability** (Section 5.1): whether changing both functional form and number of experts improves the accuracy-interpretability trade-off; (ii) **Intervenability** (Section 5.2): whether mechanism-aligned predictors respond more effectively to human interventions on concepts; and (iii) **Adaptability** (Section 5.3): whether the proposed framework can accommodate user-specified constraints without retraining the full model.

**Datasets.** We evaluate on four synthetic and five real-world datasets, covering classification and regression with both categorical and continuous concepts. *Synthetic*: **MNIST-Arithm**, a modified version of MNIST (LeCun et al., 2010) with digit pairs separated by an arithmetic operator to predict; **dSprites-Exp**, a variant of dSprites (Matthey et al., 2017) where the target is an exponential function of the object's coordinates; and **Pendulum** (Yang et al., 2020), a dataset of pendulum images with the task of predicting the pendulum's x-axis position. For real-world data, we use **MAWPS** (Koncel-Kedziorski et al., 2016), a dataset containing simple math problems in a textual form, **AWA2** (Xian et al., 2017), a classification dataset with 50 animal classes and 85 concepts, and **CUB-200** (He & Peng, 2019), a bird

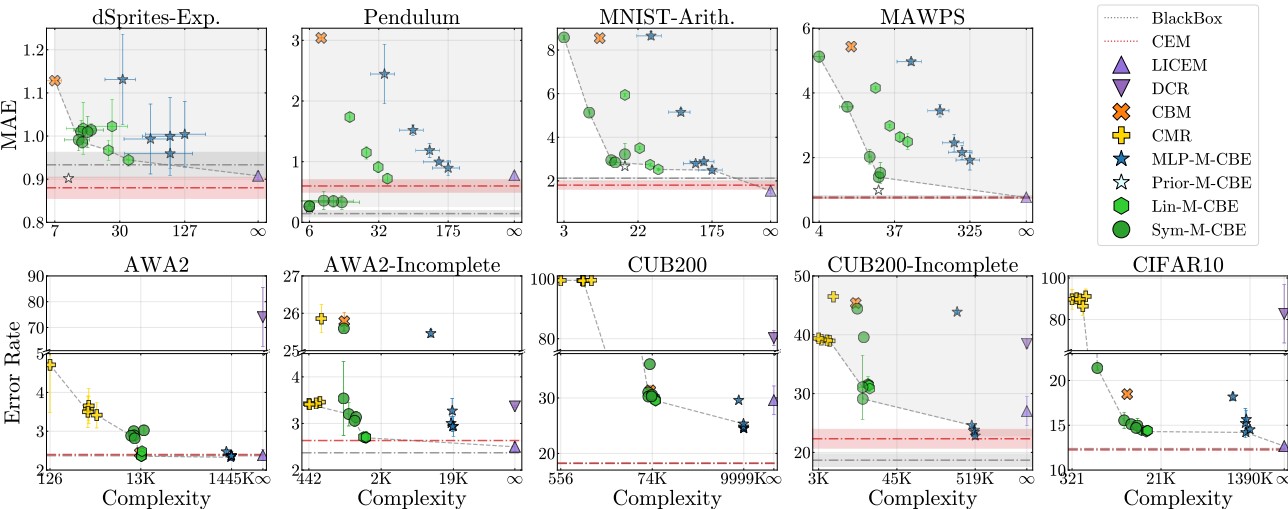

*Figure 4.* The x-axis denotes model complexity, measured as the total number of nodes across all expression trees used by the task predictor. The y-axis reports MAE for regression tasks (top) and error rate for classification tasks (bottom). For multi-expert models, multiple points are shown corresponding to different numbers of experts (1–5). The dotted curve indicates the Pareto frontier, while the shaded region marks dominated solutions. Prior-M-CBE is excluded from the Pareto frontier as it relies on ground-truth expressions. CEM and BlackBox are plotted as horizontal lines, since their label-predictor complexity is undefined (they do not operate on concept predictions). Error bars indicate 95% confidence intervals over five random seeds.

classification dataset with 200 species and 112 concepts. Following Zarlenga et al. (2025), we also evaluate incomplete versions of these datasets to study performance under missing concepts. Finally, to assess settings without concept annotations, we apply the label-free method of Oikarinen et al. (2023) to **CIFAR-10** (Krizhevsky et al., 2009). Full dataset details are provided in Appendix K.

**Baselines.** For our evaluation, we compare several concept-based architectures: **CBM** (Koh et al., 2020) (equivalent to Linear M-CBE with $M{=}1$); **CEM** (Espinosa Zarlenga et al., 2022) with a non-interpretable predictor; locally interpretable **DCR** (Barbiero et al., 2023) and **LICEM** (De Santis et al., 2025a); and globally interpretable **CMR** (Debot et al., 2024). Logic-based baselines (CMR, DCR) are omitted in regression tasks because they assume discrete concepts and outputs. We also include a standard **Blackbox** DNN as an accuracy upper-bound, and we evaluate two other instances of our framework: **Prior-M-CBE**, which uses ground-truth expressions when available, and **MLP-M-CBE**, which uses a mixture of MLPs. CMR and M-CBEs are tested with 1-5 experts. Additional details in Appendix G.

**Metrics.** For *accuracy*, we evaluate performance using the Mean Absolute Error (MAE) for regression and error rate (100−Accuracy%) for classification. For *interpretability*, guided by the principle that brevity facilitates human comprehension (Miller, 1956; Narayanan et al., 2018), and assuming individual symbols are interpretable by the user, we assess complexity via standard metrics from symbolic regression: node count, maximum depth, and the number

of variables and operators (Smits & Kotanchek, 2005). Figure 4 represents complexity as the total number of nodes in the expression tree. Further results and additional details are reported in Appendix H. When the number of expression trees is higher than one, the complexities are summed across all expression trees. For *intervenability*, we follow Koh et al. (2020); Espinosa Zarlenga et al. (2022) and measure task error as a function of the fraction of concepts replaced with ground-truth values. To evaluate the *adaptability* of the proposed model, we compare the performance of Sym-M-CBE when employing different operator sets. We note that, beyond model complexity, complementary proxies of interpretability can be derived from: intervenability, as it reveals how much and how well the task predictor leverages concepts (Koh et al., 2020); and from adaptability, since a compact expression tree is not sufficient for interpretability if it contains symbols that are unintelligible to the user.

### 5.1. Accuracy vs. Interpretability

We evaluate whether changing the number of functions, along with their complexity, improves the accuracy-interpretability trade-off. Figure 4 summarizes our findings, while detailed results, including those for other complexity metrics and concept accuracy, are provided in Appendix M.

**Exploring M-CBEs design space allows finding the best trade-offs (Figure 4).** Our results demonstrate that no single architectural choice is universally superior; rather, only exploring the M-CBEs design space guarantees finding the most accurate and interpretable model for the task at hand.

According to the dataset, we find that relaxing functional constraints with Symbolic M-CBE enables the discovery of compact, high-accuracy expressions that dominate the Pareto frontier, particularly in regression tasks (e.g., on Pendulum). Conversely, for high-dimensional classification tasks, instantiating the framework with Linear M-CBE experts provides a robust balance, offering competitive accuracy with low complexity. This highlights that explicit control over the number of experts and functional form is key for finding the best accuracy-interpretability trade-off.

**Scalability challenges for Boolean functional forms (Figure 4, bottom).** Instantiations employing Boolean expressions (e.g., recovering CMR, DCR) perform well when concept sets are small (e.g., AWA2-Incomplete and CUB200-Incomplete), offering good accuracy and very low complexity. However, they degrade substantially as the number of concepts increases, reaching near-100% error on CUB200 and CIFAR10 where concepts exceed 100. We attribute this to the rigid nature of purely conjunctive rules: when predictions rely on formulas like $c_1 \wedge \ldots \wedge c_k$, a single mispredicted concept invalidates the entire rule. This failure mode becomes increasingly critical as $K$ grows (see Appendix I for an ablation on concept size). In contrast, instantiations like Lin-M-CBE and Sym-M-CBE maintain robust performance across all concept set sizes. Moreover, these flexible forms naturally adapt to continuous concepts and targets, as demonstrated in the regression tasks.

**Black-box task predictors are generally Pareto-dominated (Figure 4).** Employing unconstrained MLPs as task predictors is rarely optimal. M-CBEs instantiations with structured functional forms (e.g., Lin-M-CBE, Sym-M-CBE) frequently match or exceed MLP accuracy on both classification and regression tasks, while being substantially less complex. Consequently, black-box predictors sporadically appear on the Pareto front, demonstrating that overparameterized predictors are suboptimal when the functional form aligns well with the underlying task.

**Multiple experts compensate for incomplete concepts and variable task logic (Figure 4).** Finally, we analyze the dimension of expert cardinality. We find that increasing the number of experts ($M > 1$) is critical in two key scenarios: first, when concept sets are incomplete (AWA2-Incomplete, CUB200-Incomplete), selecting among multiple functions recovers predictive performance; second, when the concept-to-task relationship varies across the input space (e.g., MAWPS), multiple experts allow the model to adapt to local semantic contexts. This highlights that exploring expert cardinality is also vital to ensure model performance.

**Finite number of experts allows global interpretability. (Figure 5)** Models generating input-dependent parameters (LICEM and DCR) produce a distinct expression tree for

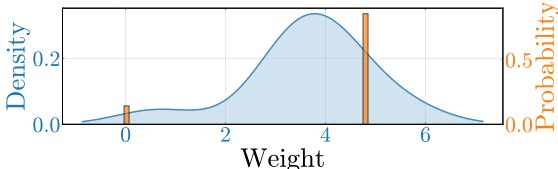

*Figure 5.* Weight distributions for the concept "has underparts color brown" toward class "Crested Auklet", comparing LICEM (blue) and Lin-M-CBE (orange, memory size $= 2$).

*Table 1.* Tree Edit Distance (TED) to the ground truth expression. In three datasets, Symbolic M-CBE recovers the exact data-generating mechanism (TED $\approx 0$).

| Model | dSprites-Exp. | Pendulum | MNIST-Arith. | MAWPS |
|---|---|---|---|---|
| Lin-M-CBE | $18.00_{\pm 0.00}$ | $5.71_{\pm 0.00}$ | $4.71_{\pm 3.23}$ | $15.00_{\pm 0.00}$ |
| MLP-M-CBE | $28.00_{\pm 0.00}$ | $43.63_{\pm 0.00}$ | $22.24_{\pm 1.75}$ | $47.00_{\pm 0.00}$ |
| Sym-M-CBE | $13.00_{\pm 0.00}$ | $0.00_{\pm 0.00}$ | $0.05_{\pm 0.09}$ | $0.00_{\pm 0.00}$ |

each sample. This implies an *infinite* model complexity, as each input corresponds to a new expression tree, making global inspection of the model infeasible. In contrast, M-CBEs learn from a *finite* set of expressions, which can be fully inspected and verified in finite time. As shown in Figure 5, LICEM produces a continuous distribution of weights around zero, reflecting its unconstrained and non-discrete parameterization. By contrast, when constrained to two parameter sets, the Linear M-CBE model selects between two discrete weight configurations, resulting in a decision process that a human can directly inspect.

## 5.2. Intervenability

CBMs allow humans to interact with the model by modifying the concept predictions. We evaluate the model's response to interventions by replacing predicted concepts with ground-truth values and by progressively increasing the fraction of corrected concepts. To simulate realistic conditions, we mildly perturb the input as $\tilde{x} = 0.9 \cdot x + 0.1 \cdot \epsilon$, where $\epsilon \sim \mathcal{N}(0, I)$. We further assess the responsiveness to interventions under stronger perturbations in Appendix J.

**Symbolic predictors align and discover true mechanisms (Figure 6, top, Table 1).** On regression tasks, Sym-M-CBE displays near-ideal intervenability: the MAE drops sharply to zero as the intervention probability approaches unity ($p_{\text{int}} \rightarrow 1$). This intervention response stems from Sym-M-CBE's ability to recover expressions that closely approximate the true concept-to-task function, (e.g., on Pendulum, $8.00 * \sin(\theta) + 10.00$). Table 1 confirms that Sym-M-CBE consistently achieves the lowest Tree Edit Distance (TED) to ground-truth expressions. The expressions learned by the methods are reported in Appendix L.

**Mixture of linear predictors excel on classification (Figure 6, bottom).** On classification tasks, Lin-M-CBE

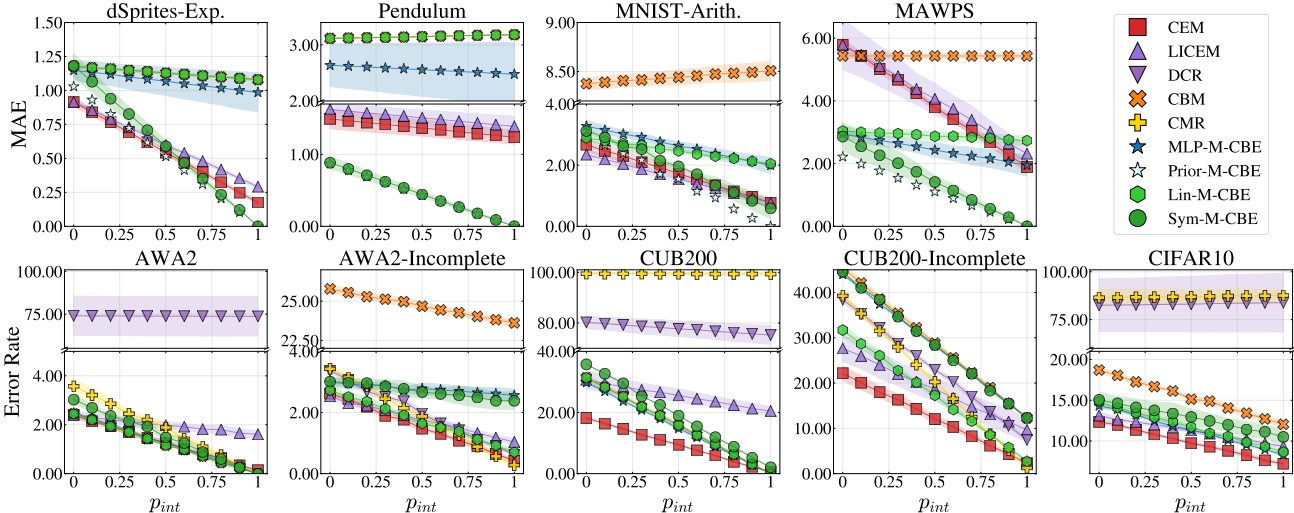

*Figure 6.* Effect of interventions on model performance. MAE (top) and error rate (bottom) as a function of intervention probability $p_{int}$. Shaded areas show 95% confidence intervals over 5 seeds. For models with multiple experts, we use the configuration at the *Pareto knee*. The knee point is identified using the maximum distance to chord method: the point with maximum perpendicular distance to the line connecting the best complexity and best accuracy extremes, representing the optimal trade-off.

*Table 2.* Performance of Sym-M-CBE across small (S), medium (M), and complete (C) operator sets, measured by MAE and model complexity ($\pm$95% confidence interval).

| Model | Pendulum | | MAWPS | |
|---|---|---|---|---|
| | MAE | Complexity | MAE | Complexity |
| MLP-M-CBE | $0.46_{\pm 0.08}$ | $706.0_{<0.01}$ | $1.05_{\pm 0.04}$ | $3592.0_{<0.01}$ |
| Sym-M-CBE (S) | $3.80_{\pm 0.02}$ | $3.0_{<0.01}$ | $2.73_{\pm 0.03}$ | $4.0_{<0.01}$ |
| Sym-M-CBE (M) | $2.12_{\pm 1.53}$ | $19.3_{\pm 5.8}$ | $1.29_{\pm 0.08}$ | $24.0_{<0.01}$ |
| Sym-M-CBE (C) | $0.46_{\pm 0.13}$ | $6.0_{<0.01}$ | $1.36_{\pm 0.21}$ | $24.0_{<0.01}$ |

emerges as the globally interpretable model most responsive to interventions. This is likely because the concept-to-task mapping in these datasets is closer to a mixture of linear functions, making Lin-M-CBE more aligned with the true underlying mapping and therefore more responsive to corrections.

### 5.3. Adaptability

While M-CBEs allow users to instantiate any interpretable functional form (e.g., linear, polynomial), Symbolic M-CBE also supports post-hoc customization *without retraining*. Once the encoder and expert selector are trained, users can modify their operator vocabulary $\mathcal{W}$ to generate tailored symbolic expressions on demand. This enables switching from expert-level trigonometric forms to student-level simple polynomials, using the same learned representations.

**Sym-M-CBE allows post-hoc adaptation to different users (Table 2).** We demonstrate this by distilling functions for three simulated user profiles: $\mathcal{W} = \{+, -\}$ (Small); $\mathcal{W} = \{+, -, \times\}$ (Medium); and an extended set including

transcendental operators (Complete). The results for *Pendulum* and *MAWPS* confirm that Sym-M-CBE adapts to these diverse constraints, demonstrating greater flexibility than other models.

## 6. Related works

Concept-based XAI (C-XAI) (Kim et al., 2018; Poeta et al., 2023) emerged to address the limited interpretability of standard attribution methods for laypeople (Rudin, 2019; Kim et al., 2023), by interpreting intermediate model representations via human-understandable concepts. Concept Bottleneck Models (CBMs) (Koh et al., 2020) extend this approach by explicitly training models to align with human semantics. Still, CBMs face significant issues: reduced accuracy compared to unrestricted models (Debot et al., 2024), limited global interpretability (Barbiero et al., 2023; De Santis et al., 2025a), and costly concept annotations (Oikarinen et al., 2023; Debole et al., 2025). Our work addresses the first two trade-offs by combining multiple task predictors with user-defined functional forms to balance accuracy and transparency, while experimentally demonstrating compatibility with label-efficient methods (Oikarinen et al., 2023).

The works most closely related to ours route samples to mixtures of interpretable predictors (Pradier et al., 2021; Ismail et al., 2022; Debot et al., 2024). However, all of these works fix the functional form a priori. Additionally, Pradier et al. (2021) and Ismail et al. (2022) apply predictors directly to raw inputs rather than concepts, while Debot et al. (2024) restricts predictors to Boolean expressions over concepts. M-CBEs extends this line of work by exposing the functional form as a flexible, user-controlled design choice, rather than

committing to a fixed form at design time.

In the context of XAI, Symbolic Regression (SR) is increasingly used to replace opaque models with explicit mathematical expressions (Dong & Zhong, 2025). These efforts fall into two categories: intrinsic approaches that embed symbolic operators directly into architectures (Sahoo et al., 2018; Biggio et al., 2021), and post-hoc approaches that approximate black-boxes with symbolic surrogates (Alaa & Van der Schaar, 2019; Bendinelli et al., 2023). While these methods effectively balance approximation error and expression complexity (Langley, 1979; Langley et al., 1981; Koza, 1994), they operate over tabular data rather than raw inputs such as images.

# 7. Conclusion

We introduced M-CBEs, a unified framework that generalizes CBMs by enabling control over two key dimensions of the task predictor: the number of experts and their functional form. Our framework subsumes existing concept-based methods while exposing a largely unexplored two-dimensional space, enabling two novel instantiations: Linear M-CBE and Symbolic M-CBE. Empirical results demonstrate that navigating this space is essential for optimal accuracy-interpretability trade-offs: algebraic forms outperform Boolean logic in high-dimensional spaces (up to +65% accuracy), multiple experts compensate for incomplete concepts, and Sym-M-CBE recovers ground-truth expressions with superior intervention responsiveness. M-CBEs establishes a principled approach for developing interpretable models that adapt to both task requirements and diverse user needs.

**Limitations.** While individual expert functions in M-CBEs are interpretable by design, the selector network $p(m \mid x)$ that routes inputs to experts operates as a black-box, limiting transparency about when and why a particular expert is selected. Additionally, Sym-M-CBE relies on heuristic search methods that are computationally intensive, particularly for large operator vocabularies or high-dimensional concept spaces. Finally, the number of experts $M$ must be specified as a hyperparameter, requiring users to balance expressiveness and interpretability through model selection.

# Acknowledgements

PB acknowledges support from the SNSF, through the project "IMAGINE" (grant ID 224226), from the Hasler Foundation, through the project "Towards Scalable Multimodal Causal Deep Learning" (grant ID 2024-05-15-70), and from the Research Foundation Flanders, through the project "Relational Concept-Based Models" (grant ID G033625N). GDF acknowledges support by the Swiss National Science Foundation (SNSF) through the grant 205121_197242 for the project "PROSELF: Semi-automated Self-Tracking Systems to Improve Personal Productivity", and by the Hasler Foundation through the project "Towards Scalable Multimodal Causal Deep Learning" (grant ID 2024-05-15-70). AC and JS acknowledge support from FFF of the University of Liechtenstein grant lbs_25_13. FG has been supported by the European Union – Horizon 2020 Program under the scheme "INFRAIA-01-2018-2019 – Integrating Activities for Advanced Communities", Grant Agreement n.871042, "SoBigData++: European Integrated Infrastructure for Social Mining and Big Data Analytics" (http://www.sobigdata.eu). This work has been partially supported by the Italian Project Fondo Italiano per la Scienza FIS00001966 "MIMOSA", and by the European Community Horizon 2020 programme under the funding schemes G.A. 101120763 "TANGO'. This work has also been supported by the EU Framework Program for Research and Innovation Horizon under the Grant Agreement No 101073307 (MSCA-DN LeMuR).

# Impact Statement

The societal implications of this work are predominantly positive. By enabling users to align model predictive mechanisms with computations they may interpret, M-CBEs democratizes access to AI across varying expertise levels, from domain specialists utilizing complex mathematical expressions to lay users favoring simplified formulations. Ethically, our framework advances responsible AI deployment by ensuring predictions remain grounded in finite, inspectable sets of interpretable expressions.

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

# A. Global Interpretability

In this appendix, we highlight the properties that M-CBEs gain by committing to a finite, fixed set of expressions, and contrast them with models whose task predictor has an interpretable form but generates a distinct expression for each input — which translates to an infinite memory of expressions in our framework — such as **LICEM** (De Santis et al., 2025a) and **DCR** (Barbiero et al., 2023). This precludes any form of global model inspection. While M-CBEs also provide instance-level explanations by routing each input to one of $M$ fixed expressions, they additionally preserve *global* interpretability through their finite expression set. This distinction leads to four concrete advantages.

- **Reasoning shortcuts**. Since models with an infinite expression memory admit many valid explanations per sample, they are not identifiable. This means they can discover equations that are mathematically valid but semantically meaningless. M-CBEs, by contrast, learn a fixed set of expressions that can be collectively evaluated for semantic consistency.

- **Verifiability**. The decision mechanisms of models with an infinite expression memory cannot be inspected without a specific input sample (Debot et al., 2024). In contrast, M-CBEs' finite set of expressions allows a human to verify all reasoning paths before deployment.

- **Knowledge Integration**. Models with an infinite expression memory lack a selector over a discrete expression set: since a different expression is produced for each input, there is no fixed formula that users can define or inject a priori. M-CBEs address this directly, as demonstrated by Prior-M-CBE, which can incorporate user-defined expressions into the fixed expression set.

- **Robustness**. An infinite expression memory increases the amount of information leaking from the input and bypassing the concept encoder. This results in worse intervention responsiveness under Out-Of-Distribution (OOD) inputs. We empirically validate this property in a dedicated experiment shown in Appendix J.

# B. Proof of Proposition 1

*Proof.* The claims are straight consequences of existing universal approximation theorems. In particular, (1) follows from the fact that an MLP (that can be represented as an expression tree with $+$ and $\times$ among operation nodes and $\sigma$ activation functions) with a single hidden layer is a universal approximator if and only if its activation function is non-polynomial (Leshno et al., 1993). On the other hand, (2) is a consequence of the classic result in mathematical analysis that any continuous function can be approximated by a piecewise linear function (Schumaker, 2007). □

# C. Proof of Proposition 2

*Proof.* The result follows from standard approximation bounds developed for numerical finite methods and interpolation methods with multivariate polynomials (Mößner & Reif, 2009). The bound is generally defined to be:

$$||f - f^\star||_\infty \leq C \cdot h^{k+1} \cdot ||f^{(k+1)}||, \tag{6}$$

where $h$ is the grid size over the input dimensions. By assuming a regular spacing so that the $M$ splines cover $M$ subspaces with equal measure over $[a, b]^n$, we can determine $h = \frac{b-a}{\sqrt[n]{M}}$. Replacing the estimation of the value of $h$ into Equation (6) completes the proof.

□

# D. Out-of-the-box M-CBEs

M-CBE characterize a wide range of CBMs architectures, including existing concept-based methods that correspond to specific inference choices within our generalized framework. In all methods discussed below, the operator set $o$ and expression tree structure $e$ are fixed by the model class; we therefore condition on $(o, e)$ and focus on how the parameters $\theta$ are modeled.

**CBM.** The original CBM (Koh et al., 2020) fixes the expression tree to a linear function over concepts and learns a single set of global parameters shared across all samples. In our framework, this corresponds to approximating $p(\theta \mid x)$ with

*Table 3.* Predictive performance (MAE) of all models on regression tasks, reported as mean $\pm$ 95% confidence interval.

| Model | dSprites-Exp. | Pendulum | MNIST-Arith. | MAWPS |
|---|---|---|---|---|
| BlackBox | $0.93_{\pm 0.03}$ | $0.14_{\pm 0.05}$ | $2.12_{\pm 0.02}$ | $0.79_{\pm 0.04}$ |
| CEM | $0.88_{\pm 0.03}$ | $0.60_{\pm 0.10}$ | $1.80_{\pm 0.18}$ | $0.73_{\pm 0.02}$ |
| LICEM | $0.90_{\pm 0.01}$ | $0.77_{\pm 0.07}$ | $1.55_{\pm 0.11}$ | $0.74_{\pm 0.03}$ |
| MLP-M-CBE | $1.06_{\pm 0.05}$ | $2.30_{\pm 0.46}$ | $2.81_{\pm 0.14}$ | $1.95_{\pm 0.23}$ |
| Prior-M-CBE | $0.90_{\pm 0.01}$ | $0.24_{\pm 0.01}$ | $2.67_{\pm 0.11}$ | $1.00_{\pm 0.01}$ |
| Kan-M-CBE | $0.91_{\pm 0.01}$ | $0.24_{\pm 0.02}$ | $4.47_{\pm 1.26}$ | $0.96_{\pm 0.02}$ |
| Lin-M-CBE | $1.13_{\pm 0.01}$ | $3.04_{\pm 0.00}$ | $2.74_{\pm 0.05}$ | $2.63_{\pm 0.12}$ |
| Sym-M-CBE | $1.02_{\pm 0.01}$ | $0.33_{\pm 0.12}$ | $2.92_{\pm 0.16}$ | $1.33_{\pm 0.18}$ |

a delta distribution that places all its mass on a single, input-independent parameter value $\theta$, where $(o, e)$ define a linear expression (i.e., $o = \{+, \times\}$ with appropriate edges). We omit $v$ to simplify the notation since all available concepts are used. The conditional distribution induced by the CBM can be expressed as $p(y, c \mid x; o, e) = p(y \mid c; o, e, \theta)\, p(c \mid x)$. For a regression task, the task predictor is modeled as a Gaussian: $p(y \mid c; o, e, \theta) = \mathcal{N}\big(y; f_{(o,e)}(c; \theta), \sigma^2\big)$, where the mean is a linear function of the concepts: $f_{(o,e)}(c; \theta) = \theta^\top c + b$, with $\theta = w$ representing the weights on the concepts, $b$ the bias term, and $\sigma^2$ denoting the variance of the Gaussian noise.

**CMR.** CMR (Debot et al., 2024) instead constrains the expression tree to Boolean formulas and restricts parameters to a finite discrete set. Each concept can appear positively $(+1)$, negated $(-1)$, or be absent $(0)$ from the formula. CMR learns a memory of $M$ parameter configurations $\{\theta_1, \ldots, \theta_M\}$ and models $p(\theta \mid x) = \sum_{m=1}^{M} p(m \mid x)\, p(\theta \mid m)$.

In the limiting case $M \to \infty$, the distribution is no longer discretized but instead produces sample-specific parameters. Both **LICEM** (De Santis et al., 2025a) and **DCR** (Barbiero et al., 2023) correspond to this limiting case, differing only in the structure enforced by $(o, e)$. LICEM constrains the expression tree's structure to represent a linear equation, while DCR constrains it to represent a Boolean expression.

# E. Symbolic predictor ablation

We compare Sym-M-CBE with Kan-M-CBE, an alternative instantiation that uses Kolmogorov-Arnold Networks (Liu et al., 2025).

Table 3 shows that both models achieve comparable MAE, with Kan-M-CBE obtaining slightly lower errors on all datasets except MNIST-Arith.

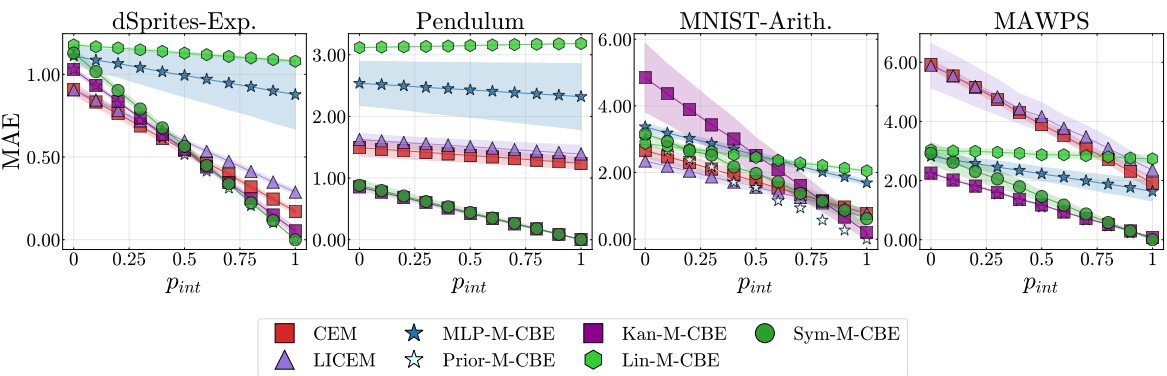

*Figure 7.* Effect of interventions on model performance. MAE as a function of intervention probability $p_{int}$. Shaded areas show 95% confidence intervals over 5 seeds.

Figure 7 demonstrates that Kan-M-CBE exhibits strong responsiveness to interventions, achieving MAE values at $p_{int} \approx 1.0$ comparable to Prior-M-CBE.

However, despite achieving competitive predictive accuracy and intervention responsiveness, Kan-M-CBE produces expressions with substantially higher complexity (Table 4) compared to other methods. This stems from the KAN training procedure: the model first learns network activations (splines), then applies sparsification, and finally adds affine parameters

*Table 4.* Complexity (Node Count) of learned expressions, reported as mean $\pm$ 95% confidence interval.

| Model | dSprites-Exp. | Pendulum | MNIST-Arith. | MAWPS |
|---|---|---|---|---|
| MLP-M-CBE | $41.6_{\pm 8.6}$ | $41.6_{\pm 8.6}$ | $184.0_{\pm 0.0}$ | $310.4_{\pm 65.9}$ |
| Prior-M-CBE | $10.0_{\pm 0.0}$ | $6.0_{\pm 0.0}$ | $16.0_{\pm 0.0}$ | $24.0_{\pm 0.0}$ |
| Kan-M-CBE | $113.8_{\pm 14.7}$ | $101.6_{\pm 15.2}$ | $267.0_{\pm 35.4}$ | $891.2_{\pm 41.6}$ |
| Lin-M-CBE | $7.4_{\pm 1.2}$ | $8.0_{\pm 0.0}$ | $32.0_{\pm 0.0}$ | $44.0_{\pm 0.0}$ |
| Sym-M-CBE | $13.2_{\pm 2.0}$ | $10.6_{\pm 6.7}$ | $12.0_{\pm 2.0}$ | $25.0_{\pm 2.0}$ |

to each spline (e.g., transforming an activation $g(x)$ into $a \cdot g(b \cdot x + c) + d$), resulting in considerably larger expressions. The compactness of expressions extracted from KAN networks depends critically on multiple hyperparameters, including entropy regularization for sparsification, pruning strength, and the symbolic substitution process that replaces activations with mathematical symbols. This increased complexity makes Kan-M-CBE more difficult to tune in practice while yielding inferior performance in terms of expression size and, ultimately, alignment with ground-truth mechanisms, as shown in Table 5. We emphasize that this does not imply KANs are inherently inferior; rather, we found them more challenging to train and tune compared to the genetic programming-based symbolic regression approach implemented in PySR (Cranmer, 2023).

*Table 5.* TED between learned and ground truth expressions, reported as mean $\pm$ 95% confidence interval.

| Model | dSprites-Exp. | Pendulum | MNIST-Arith. | MAWPS |
|---|---|---|---|---|
| MLP-M-CBE | $45.45_{\pm 8.55}$ | $39.02_{\pm 8.66}$ | $43.92_{\pm 0.08}$ | $80.60_{\pm 16.46}$ |
| Prior-M-CBE | $0.00_{\pm 0.00}$ | $0.00_{\pm 0.00}$ | $0.00_{\pm 0.00}$ | $0.00_{\pm 0.00}$ |
| Lin-M-CBE | $17.40_{\pm 1.18}$ | $5.71_{\pm 0.00}$ | $4.70_{\pm 0.01}$ | $15.00_{\pm 0.00}$ |
| Sym-M-CBE | $9.85_{\pm 8.19}$ | $7.02_{\pm 10.90}$ | $0.05_{\pm 0.02}$ | $0.70_{\pm 1.37}$ |

# F. Training details

All model variants share a common training framework implemented in PyTorch Lightning. We use *AdamW* as the optimizer with dataset-specific learning rates (ranging from $10^{-4}$ to $10^{-1}$) and a *ReduceLROnPlateau* scheduler that decreases the learning rate by a factor of $\gamma = 0.5$ when the validation loss fails to improve for 25 consecutive epochs. Early stopping is applied with a patience of 50. Training proceeds for a maximum of 600 epochs. To minimize leakage (Marconato et al., 2022), all the concept-based methodologies are trained in a disjoint manner (Koh et al., 2020): the task predictor uses ground-truth concept labels during training. In addition to that, when the concepts are binary, we apply hard thresholding.

The total loss is a weighted combination of concept and task prediction losses:

$$\mathcal{L}_{\text{total}} = \lambda_c \mathcal{L}_{\text{concept}} + \lambda_y \mathcal{L}_{\text{task}} \tag{7}$$

where $\lambda_c = 1.0$ and $\lambda_y = 0.1$ across all models. For classification tasks, we use binary cross-entropy for concept prediction and cross-entropy for task prediction. For regression tasks, mean squared error is employed for both. In all methodologies, sparsity is promoted by tuning the respective hyperparameters in order to maximize the accuracy-interpretability trade-off. Nevertheless, for the methodologies we proposed, we followed a multi-stage pipeline.

For all the methodologies using a mixture of experts, we employ a selector implemented as an MLP with output size equal to the number of experts. During training, we employ a *Gumbel-Softmax* (Jang et al., 2016) to sample an index according to the distribution produced by the selector $p(m \mid x)$ for the specific sample. We gradually reduce the temperature $\tau$ of the *Gumbel-Softmax* from $\tau = 2$ to $\tau = 0.05$ following a cosine decay. This schedule ensures that the selection distribution becomes increasingly peaked as training progresses. At test time, a single expression index $m$ is sampled from $p(m \mid x)$, and the prediction is computed using only the selected expression.

**Lin-M-CBE.** Training proceeds in two stages. The first stage trains all components (concept encoder, selector, and linear memory) end-to-end with $\ell_1$ regularization on weight matrices ($\lambda_1 = 10^{-5}$) and $\ell_2$ regularization on bias terms. Upon initial convergence, the second stage applies hard thresholding: weights with absolute values below $\tau = 10^{-6}$ are set to zero and frozen. The remaining non-zero parameters are then fine-tuned for up to 600 additional epochs, promoting sparse, more interpretable linear equations.

**Sym-M-CBE.** Training follows a three-stage pipeline. In the first stage, the complete model is trained with black-box neural network predictors (MLPs) using the shared training configuration. Upon convergence, the symbolic regression phase begins: predictions are collected from the trained model on the entire training set. PySR then discovers symbolic equations for each placeholder blackbox independently, fitting equations using the subset of data points selected by the selector for that specific placeholder blackbox. The symbolic regression search uses the following configuration: 40 populations of size 60 evolve for 100 iterations with 380 cycles per iteration. For classification tasks, operators are restricted to $\{+, -, \times\}$ with maximum equation complexity $5 \times k$ (where $k$ is the number of concepts). For regression tasks, the operator set is expanded to include $\{\sin, \cos, \exp, \log, \tan, \tanh, x^2, x^3, \sqrt{x}, x^{-1}\}$ with maximum complexity 40. Discovered equations are substituted into the model as symbolic predictor modules with trainable numeric parameters (exponents remain fixed).

In the third stage, the entire model is fine-tuned with the symbolic predictor for up to 600 epochs. To account for the reduced parameter count, the learning rate is increased by a factor of 5 relative to the initial training phase. Only the numeric parameters in the symbolic equations and the upstream networks (concept encoder and selector) are trainable; the structural form of the equations remains fixed.

# G. Implementation details

## G.1. Baselines

This section provides implementation details for all methods. We implemented all concept-based baselines using the PyC library (Barbiero et al., 2025). To ensure computational efficiency, all models operate on pre-computed embeddings rather than raw inputs. We employ pre-trained backbones: `facebook/dinov2-base` for high-resolution images, `ResNet18` for low-resolution images, and `google/flan-t5-large` for textual data. Unless otherwise specified, all networks use LeakyReLU activations with a default hidden dimensionality of 64 (adjusted per dataset). Complete hyperparameter configurations are provided in the supplementary materials.

The **Blackbox** baseline is a single hidden-layer MLP mapping embeddings directly to task predictions. **CBM** (Koh et al., 2020) follows the standard architecture with a linear task predictor. **CEM** (Espinosa Zarlenga et al., 2022) uses concept embeddings of dimensionality 16 with an MLP as task predictor. **DCR** (Barbiero et al., 2023), **LICEM** (De Santis et al., 2025a), and **CMR** (Debot et al., 2024) all use concept embeddings of dimensionality 16. Both CEM and LICEM are adapted to continuous concepts following Ismail et al. (2024). Specifically, in order to preserve responsiveness to interventions, the concept embedding $\hat{c}$ of each concept is multiplied by the respective concept prediction.

## G.2. Proposed methods

We implement three variants of our M-CBEs framework, each employing different symbolic reasoning strategies over learned concept representations.

All variants share a common architecture consisting of: (i) a concept encoder, (ii) a selector network that produces a probability distribution over $M$ expressions (experts), and (iii) a task predictor that executes the selected expression. The models differ primarily in how the expressions are obtained and parameterized.

**Prior-M-CBE.** Symbolic equations are provided as SymPy expressions. Each memory slot contains a fixed equation $f_m$ whose structure and parameters remain frozen.

**MLP-M-CBE** In MLP-M-CBE each expert is an MLP with 1 hidden layer with hidden size set as specified in Section G.1. We apply an L1 regularization to the parameters of each MLP. This loss term is multiplied by $1e - 5$ and added to the loss.

**Kan-M-CBE.** In Kan-M-CBE, each expert in the mixture is implemented using a KAN. The architecture is specified by a width vector $\mathbf{w}$, which varies based on the task. We use a deeper architecture: $\mathbf{w} = [n_c, n_c + 1, n_c + 1, 1]$, where $n_c$ is the number of concepts. Each edge $(i, j)$ between layers is parameterized by a univariate B-spline function $\phi_{i,j} : \mathbb{R} \to \mathbb{R}$ with 5 grid points and cubic basis functions ($k = 3$). KANs inherently learn smooth, interpretable functions and support automatic symbolic conversion. During training, we apply KAN-specific regularization to encourage sparsity. We set the hyperparameter related to this sparsity to $\lambda_{\text{sparsity}} = 0.001$. After the first training, each KAN expert is pruned and each spline is substitute with the symbol in $\{+, -, \times, \sin, \cos, \exp, \log, \tan, \tanh, x^2, x^3, \sqrt{x}, x^{-1}, x^{-2}\}$ the best fit the spline

**Lin-M-CBE.** Each memory slot stores a weight matrix $W_m$ and bias vector $b_m$. We apply $\ell_1$ regularization on weights with coefficient $\lambda_1 = 10^{-5}$ and hard thresholding with $\tau = 10^{-6}$ after initial training.

**Sym-M-CBE.** We use PySR (Cranmer, 2023) with 40 populations of size 60, 100 iterations, and 380 cycles per iteration. For classification: operators $\{+, -, \times\}$, maximum complexity $5 \times k$ (where $k$ is the number of concepts). For regression: additional operators $\{\sin, \cos, \exp, \log, \tan, \tanh, x^2, x^3, \sqrt{x}, x^{-1}, x^{-2}\}$, maximum complexity 40. Discovered equations are converted to SymPy expressions with trainable parameters.

## H. Complexity metrics

To evaluate the complexity of the symbolic expressions discovered, we employ several metrics that quantify different structural properties of the expression trees. Using the notation from Section 3.1, we additionally denote by $T_n$ the subtree rooted at node $n \in N$, by $N_{T_n}$ its node set, and by $d(n)$ the depth of $n$ (i.e., the path length from the root to $n$). For multi-mechanism predictors with $M$ expression trees $\{T^{(1)}, \ldots, T^{(M)}\}$, we report aggregate metrics summed across all trees. Using this notation, the complexity metrics are defined as follows:

- **Node count**: The total number of elements in the expression tree: $\text{NodeCount}(T) = |N| = |V| + |O| + |\Theta|$. This is the default complexity metric employed in the PySR library (Cranmer, 2023).

- **Tree depth**: The maximum nesting level of operations, reflecting the hierarchical complexity of the expression: $\text{Depth}(T) = \max_{n \in N} d(n)$.

- **Expression complexity**: The sum of subtree sizes across all nodes, which penalizes deeply nested structures more heavily than shallow ones (Keijzer & Foster, 2007; Smits & Kotanchek, 2005): $\text{ExprComplexity}(T) = \sum_{n \in N} |N_{T_n}|$.

- **Total variables**: The number of unique concept variables appearing in the expression: $\text{TotalVars}(T) = |V|$. Note that the same concept $c_k$ may appear multiple times in the tree; this metric counts unique variables, not occurrences.

- **Total operations**: The count of operator nodes in the tree: $\text{TotalOps}(T) = |O|$. Since operators correspond to internal (non-leaf) nodes, this equals the number of non-terminal nodes in $T$.

- **Weighted node count**: A variant of node count that assigns different weights to operators based on their complexity. Let $w : \mathcal{O} \to \mathbb{R}^+$ be a weight function, where basic arithmetic operators $\{+, -, \times, \div\}$ receive unit weight $w(o) = 1$, while transcendental functions (e.g., sin, cos, exp, log) receive weight $w(o) = 2$. Variables and constants receive unit weight. The weighted node count is $\text{WeightedCount}(T) = |V| + |\Theta| + \sum_{o \in O} w(o)$. This metric favors expressions composed of simpler primitives.

## I. Concept Size ablation

In this section, we investigate why Boolean functional forms fail to scale with the number of concepts. To systematically study this phenomenon, we evaluate Lin-M-CBE and concept-based baselines (CBM, CMR, DCR, LICEM) on CUB200 and CIFAR10 datasets, both of which feature concept bottlenecks with more than 100 concepts. We systematically vary the bottleneck size by randomly subsampling different subsets of concepts from the original set. For each bottleneck size, we train all models on the resulting modified dataset and evaluate their performance. This process is repeated across multiple bottleneck sizes to observe how model accuracy changes as a function of the number of available concepts. For Lin-M-CBE and CMR, we set the number of experts to 2.

As shown in Figure 8, models employing Boolean functional forms (CMR, DCR) exhibit a clear degradation in accuracy as the concept bottleneck size increases. This behavior strengthens our hypothesis that conjunctive Boolean rules become increasingly brittle with larger concept sets: the probability of mispredicting at least one concept grows with dimensionality, causing the entire logical formula to fail. In contrast, models with more flexible functional forms (Lin-M-CBE, CBM, LICEM) demonstrate improved accuracy as the bottleneck size increases. These architectures benefit from the additional task-relevant information provided by larger concept sets, as their continuous aggregation mechanisms are more robust to individual concept prediction errors.

## J. Interventions under Noisy Inputs

One advantage of intervenable models is that they can be actively corrected when deployed in real-world environments subject to OOD shifts. Nevertheless, recent works (Zarlenga et al., 2025; De Santis et al., 2025b) have shown that concept-based architectures that allow residual input information to bypass the concept bottleneck suffer a significant degradation

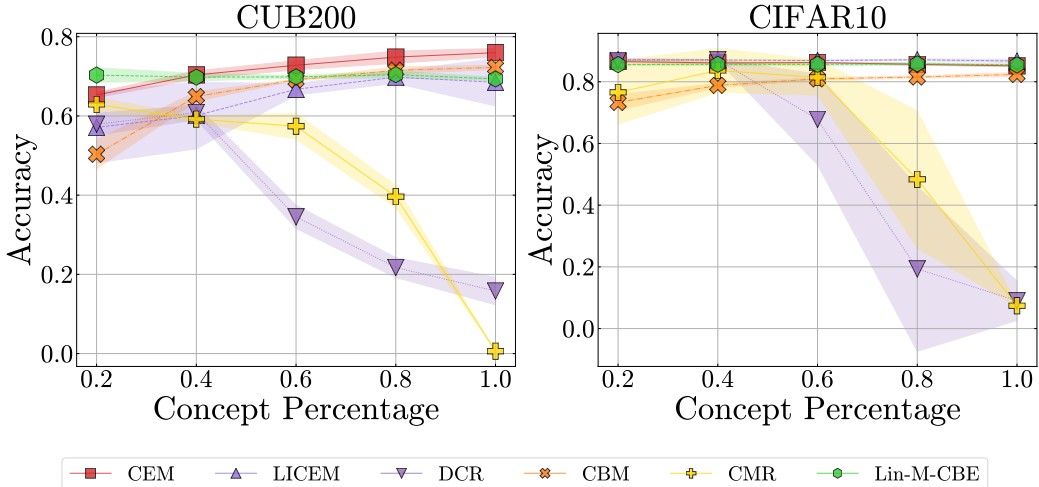

*Figure 8.* Effect of concept bottleneck size on model accuracy for CUB200 (left) and CIFAR10 (right). The x-axis shows the percentage of selected concepts, and the y-axis reports accuracy. For task predictors with Boolean functional forms (CMR, DCR), accuracy decreases as more concepts are included, while models with flexible functional forms (Lin-M-CBE, Sym-M-CBE) improve as bottleneck size increases. Error bars indicate 95% confidence intervals over 5 seeds.

in terms of responsiveness to interventions under OOD inputs — i.e., they cannot recover predictive accuracy on the downstream task even if we replace the concept predictions with the corresponding ground-truth values.

In this appendix, we assess intervention robustness under a noisy input regime on all regression datasets (MNIST-Arithm, dSprites-Exp, Pendulum, MAWPS), as these are the only datasets for which the true concept-to-task mechanisms are known, allowing us to understand whether a task predictor whose functional form is closer to the true concept-to-task mapping is also more responsive to interventions in OOD settings. Since CMR and DCR assume discrete concepts and outputs, they are inapplicable to regression tasks and are therefore excluded; results are shown for all remaining models. Specifically, we corrupt the input $x$ as $\tilde{x} = 0.5 \cdot x + 0.5 \cdot \epsilon$, where $\epsilon \sim \mathcal{N}(0, I)$. As half of the information in $\tilde{x}$ is pure Gaussian noise, this provides a challenging setting in which the concept encoder must operate on heavily degraded inputs. We then measure task performance as ground-truth interventions are progressively applied, following the same protocol as in Section 5.2.

Figure 9 reveals how both the functional form and the number of experts affect intervention responsiveness. With respect to functional form, aligning the task predictor with the true underlying concept-to-task mechanism enables the model to recover predictive performance through ground-truth interventions, as demonstrated by Prior-M-CBE and Sym-M-CBE. A finite set of experts, on the other hand, limits the amount of input leakage that can bypass the concept bottleneck. This leakage is unconstrained in models with continuous parameterization such as LICEM, which exhibits a high MAE ($\approx 100$) in the absence of interventions and struggles to recover predictive performance even when all concepts are corrected ($p_{\text{int}} = 1$).

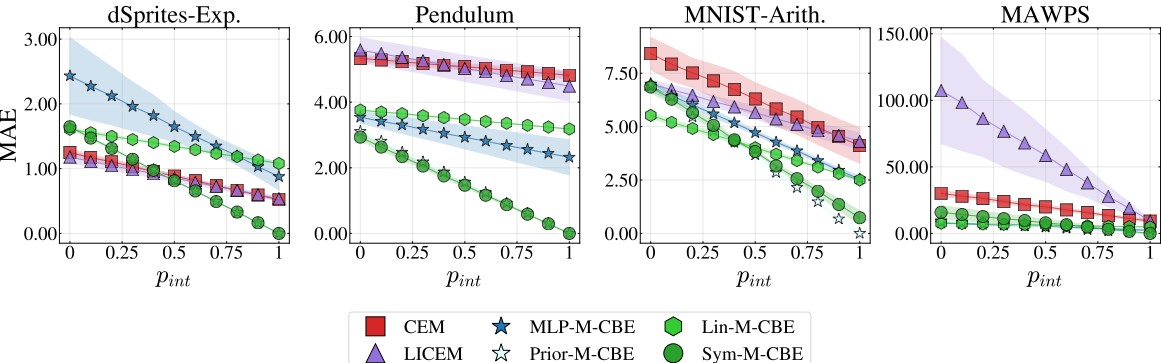

*Figure 9.* Effect of interventions on model performance under heavily noisy inputs ($\tilde{x} = 0.5 \cdot x + 0.5 \cdot \epsilon$, $\epsilon \sim \mathcal{N}(0, I)$). MAE as a function of intervention probability $p_{\text{int}}$ across all regression datasets. Shaded areas show 95% confidence intervals over 5 seeds. For models with multiple experts, the number of experts is set to match the true number of concept-to-task mechanisms in each dataset.

# K. Datasets details

## K.1. Synthetic Datasets

**MNIST-Arithm** The MNIST dataset (LeCun et al., 2010) is a widely used benchmark consisting of grayscale images of handwritten digits. It contains 60,000 training examples and 10,000 test examples, sampled from the same distribution, with each image annotated by its corresponding digit label. MNIST-Arithm is derived from MNIST through the following procedure. First, we specify the total number of images to generate (in our experiments, 100000). Each generated image is created by randomly sampling two images from MNIST, extracting their digit labels, and combining them into a new image separated by an arithmetic operation (addition, subtraction, multiplication, or division). Correspondingly, each image is annotated with: (i) two concept variables representing the digits contained in the image, and (ii) a task variable corresponding to the result of the arithmetic operation. Then, the resulting dataset is split into training (70%), validation (10%), and test (20%) sets. Finally, each image is preprocessed using a pre-trained `facebook/dinov2-base` model with default Hugging Face weights.

**dSprites-Exp** The dSprites dataset (Matthey et al., 2017) is a widely used dataset containing 737280 images of 2D white shapes on a black background generated from 6 ground truth independent latent factors: color, shape, scale, rotation, and x and y positions of a sprite. dSprites-Exp dataset is derived from dSprites through the following procedure. First, we specify the total number of images to generate (in our experiments, 100,000). Each generated image is sampled from the original dataset and it is labeled with original latent factors, which serve as our concepts, except for color, which is omitted since it is constant ("white"). Next, a target variable is defined for each sample as $\text{target} = \exp\left(\sin(2\pi\,\text{x\_position}) + \cos(2\pi\,\text{y\_position})\right)$. Then, the resulting dataset is split into training (70%), validation (10%), and test (20%) sets. Finally, all images are preprocessed using a pre-trained `facebook/dinov2-base` model with default weights from Hugging Face.

**Pendulum** Pendulum is a synthetic dataset originally introduced in Yang et al. (2020). It consists of ∼7k generated images of a swinging pendulum with a moving light source. The positions of the illumination source and the pendulum angle relative to the vertical determine the position and length of the pendulum's shadow on a horizontal plane. We consider a subset of 100000 images from the original dataset, obtained by sampling 100 pendulum angles ranging from $-200°$ to $200°$ and 1000 light source angles between $60°$ and $140°$ (where the light source angle is defined as the angle between the line connecting the pendulum's center of rotation to the center of the light bulb and the pendulum's vertical line). As concepts, we use the radiant representation of the angles, and as a target, we consider the $x$-position of the pendulum ball, to avoid overly complex ground-truth mechanisms. The dataset is then split into training (70%), validation (10%), and test (20%) sets. All images are finally preprocessed using the pre-trained `facebook/dinov2-base` model with default weights from Hugging Face.

## K.2. Real-world Datasets

**AWA2.** This dataset is the Animals with Attributes 2 dataset (Xian et al., 2017), consisting of RGB images depicting one of 50 animal species. Each image is annotated with a species label and 85 numeric attributes, which we treat as concept labels, while the species serves as the target in our classification problem. Following prior work (Alaa & Van der Schaar, 2019), we generate train-validation-test splits using a random 60%–20%–20% partition. During training, samples are randomly cropped and flipped. Finally, all images are preprocessed using the pre-trained `facebook/dinov2-base` model with default weights from Hugging Face.

**AWA2-Incomplete.** This dataset is derived from Animals with Attributes 2 (Xian et al., 2017), following the same procedure as AWA2, with the only difference that we consider only a subset of the 85 concepts. Specifically, we retain the following concepts: "black", "gray", "stripes", "hairless", "flippers", "paws", "plains", "fierce", "solitary".

**CUB-200.** This dataset is the Caltech-UCSD Birds-200-2011 dataset (He & Peng, 2019). Specifically, each sample consists of an RGB image of a bird annotated with one of 200 species and 312 binary attributes. In our experiments, we adopt the 112 bird attributes selected in (Koh et al., 2020) as binary concept annotations and use bird species as the downstream classification task. All images are preprocessed, and the dataset is then split following the same procedure described in Espinosa Zarlenga et al. (2022). Finally, all images are encoded using the pre-trained `facebook/dinov2-base` model with the default Hugging Face weights.

**CUB-200-Incomplete.** This dataset is a subset of CUB-200 where we select the following concepts: "has_bill_shape", "has_head_pattern", "has_breast_color", "has_bill_length", "has_wing_shape", "has_tail_pattern", "has_bill_color". **CIFAR-10.**

The original dataset (Krizhevsky et al., 2009) contains 60,000 RGB images, each belonging to one of 10 object categories. Following prior work of Oikarinen et al. (2023), we annotate each sample with a set of binary concepts. We adopt the original train-test split, reserving $10\%$ of the training set for validation. Finally, all images are encoded using the pre-trained `google/vit-base-patch32-224-in21k` model with the default Hugging Face weights.

**MAWPS.** Is a benchmark dataset for evaluating models on *math word problem solving* (Koncel-Kedziorski et al., 2016). It contains $\sim 3.3K$ English arithmetic and simple algebra problems, each annotated with a ground-truth equation and solution. We filter out samples whose concept-to-task expression is too rare, retaining only those belonging to the 4 most frequent expression templates. To increase the size of the training split, we augment it by replacing the numeric values (digits) appearing in each mathematical formula with new random values, and computing the corresponding task label by evaluating the formula over the augmented concept values.

## L. Learned Expressions

This section presents the symbolic expressions learned by Lin-M-CBE and Sym-M-CBE across experimental datasets. For regression tasks (Table 6), we set the number of experts $M$ equal to the number of underlying mechanisms mapping concepts to task outputs.

For classification tasks (Table 7), due to space constraints, we only report results for CUB200-Incomplete and AWA2-Incomplete, as these datasets have a reduced number of concepts, making the expressions more compact and interpretable. These expressions provide concrete examples of how M-CBEs instantiations translate concept predictions into task predictions.

*Table 6.* Equations learned by the proposed instantiations on the regression datasets.

| Dataset | Model | Equations |
|---|---|---|
| dSprites-Exp. | Lin-M-CBE | $-2.5430 * value\_x\_position - 0.0001 * value\_y\_position + 2.9265$ |
| dSprites-Exp. | Prior-M-CBE | $exp(sin(6.28000020980835 * value\_x\_position) + cos(6.28000020980835 * value\_y\_position))$ |
| dSprites-Exp. | Sym-M-CBE | $exp(sin(6.28318309783936 * value\_x\_position) + cos(6.28318452835083 * value\_y\_position))$ |
| Pendulum | Lin-M-CBE | $2.3583 * theta - 0.0042 * phi + 10.0064$ |
| Pendulum | Prior-M-CBE | $8.0 * sin(theta) + 10.0$ |
| Pendulum | Sym-M-CBE | $8.00006008148193 * sin(theta) + 9.9999361038208$ |
| MNIST-Arith. | Lin-M-CBE | $3.7693 * first\_digit + 3.7327 * second\_digit - 1.0437$
$0.6945 * first\_digit - 0.5778 * second\_digit + 0.3109$
$1.0767 * first\_digit + 1.0075 * second\_digit - 0.2216$
$2.6496 * first\_digit + 2.6543 * second\_digit - 1.3471$ |
| MNIST-Arith. | Prior-M-CBE | $first\_digit * second\_digit$
$first\_digit / second\_digit$
$first\_digit + second\_digit$
$first\_digit - second\_digit$ |
| MNIST-Arith. | Sym-M-CBE | $first\_digit * second\_digit$
$first\_digit / second\_digit$
$first\_digit + second\_digit$
$first\_digit - 0.834531188011169 * second\_digit$ |
| MAWPS | Lin-M-CBE | $-0.9211 * N\_00 + 0.1138 * N\_01 + 0.1780 * N\_02 - 0.1387$
$-4.3256 * N\_00 + 0.0659 * N\_01 - 0.0764 * N\_02 - 0.2263$
$1.2093 * N\_00 - 0.0052 * N\_01 + 0.0454 * N\_02 - 0.0351$
$4.5133 * N\_00 - 0.1206 * N\_01 - 0.0568 * N\_02 + 0.2450$ |
| MAWPS | Prior-M-CBE | $N\_00 * (N\_01 - 1.0 * N\_02)$
$N\_00 * (N\_01 + N\_02)$
$N\_02 * (N\_00 + N\_01)$
$N\_02 * (N\_00 - 1.0 * N\_01)$ |
| MAWPS | Sym-M-CBE | $N\_00 * (N\_01 - 0.999999642372131 * N\_02)$
$N\_00 * (N\_01 + N\_02)$
$N\_02 * (N\_00 + N\_01)$
$N\_02 * (N\_00 - 1.00000178813934 * N\_01)$ |

## M. Detailed Results

### M.1. Task accuracy and complexity

In this section, we provide detailed experimental results (Tables 8 to 12) showing the accuracy, MAE, and all complexity metrics for the various methods across different datasets. Additionally, since there are as many Pareto frontiers as there are complexity metrics, each table includes a column representing the number of times a model appeared on the Pareto frontier.

*Table 7.* Learned expressions for classification tasks. For each dataset, we randomly select three classes and report the learned expression mapping concept predictions to that class.

| Dataset | Selected class | Model | Explanation |
|---|---|---|---|
| AWA2_incomplete | Squirrel | Lin | $-8.3725 * black + 4.4533 * gray - 1.2107 * stripes - 5.6324 * hairless - 1.7003 * flippers + 1.8803 * paws - 3.8328 * plains - 4.6689 * fierce + 2.3280 * solitary + 0.4597$ |
| AWA2_incomplete | Squirrel | Sym | $-2.33258175849915 * black - 2.04653811454773 * fierce - 1.71986174583435 * flippers + 1.87418258190155 * gray - 2.48390746116638 * hairless + 1.33075654506683 * paws - 2.27515959739685 * plains + 1.38696956634521 * solitary - 1.98490846157074 * stripes + 0.515300989151001$ |
| AWA2_incomplete | Deer | Lin | $-5.5206 * black - 2.8409 * gray - 1.9280 * stripes - 1.6717 * hairless - 1.6070 * flippers - 2.7865 * paws + 4.2462 * plains - 2.6639 * fierce - 3.7920 * solitary + 4.0240$ |
| AWA2_incomplete | Deer | Sym | $-2.21854496002197 * black - 1.93461465835571 * fierce - 1.4765248298645 * flippers - 1.88623571395874 * gray - 1.58335506916046 * hairless - 1.48845815658569 * paws + 2.08511114120483 * plains - 1.87786483764648 * solitary - 2.1214337348938 * stripes + 2.2183256149292$ |
| AWA2_incomplete | Rabbit | Lin | $3.5934 * black + 0.4045 * gray - 2.0988 * stripes - 1.6350 * hairless - 1.6533 * flippers + 6.4211 * paws + 2.5394 * plains - 6.1008 * fierce - 8.1255 * solitary - 3.5315$ |
| AWA2_incomplete | Rabbit | Sym | $1.18550539016724 * black - 1.97509169578552 * fierce - 1.74324333667755 * flippers + 1.56745767593384 * gray - 0.0243255645036697 * gray - 1.87004804611206 * hairless + 2.10452556610107 * paws + 1.38752210140228 * plains - 2.72675681114197 * solitary - 2.14433598518372 * stripes - 1.12651097774506$ |
| CUB200_incomplete | Laysan_Albatross | Lin | $-1.6006 * has\_bill\_shape\_curved\_up\_or\_down + 5.3314 * has\_bill\_shape\_dagger - 1.6190 * has\_bill\_shape\_hooked - 1.8209 * has\_bill\_shape\_needle - 0.7035 * has\_upperparts\_color\_red - 3.4107 * has\_upperparts\_color\_buff - 1.2424 * has\_underparts\_color\_blue - 1.2530 * has\_underparts\_color\_brown - 1.2431 * has\_underparts\_color\_iridescent - 1.7109 * has\_underparts\_color\_purple + 0.6020 * has\_underparts\_color\_rufous - 1.2875 * has\_underparts\_color\_grey + 1.3830 * has\_underparts\_color\_white - 2.5163 * has\_underparts\_color\_red - 1.8923 * has\_tail\_shape\_fan\_shaped\_tail - 1.2673 * has\_tail\_shape\_pointed\_tail - 2.0584 * has\_upper\_tail\_color\_olive - 1.5512 * has\_upper\_tail\_color\_green - 2.2977 * has\_upper\_tail\_color\_pink - 1.1097 * has\_head\_pattern\_eyebrow - 2.8353 * has\_head\_pattern\_eyering + 9.5280 * has\_head\_pattern\_plain - 0.4318$ |
| CUB200_incomplete | Laysan_Albatross | Sym | $-2.12568759918213 * has\_bill\_shape\_curved\_up\_or\_down + 1.16855323314667 * has\_bill\_shape\_dagger - 1.65060842037201 * has\_bill\_shape\_hooked - 1.96898102760315 * has\_bill\_shape\_needle - 1.50070405006409 * has\_head\_pattern\_eyebrow - 2.59437537193298 * has\_head\_pattern\_eyering + 1.62642562389374 * has\_head\_pattern\_plain - 1.6626957654953 * has\_tail\_shape\_fan\_shaped\_tail - 2.22473764419556 * has\_tail\_shape\_pointed\_tail - 2.65573024749756 * has\_underparts\_color\_blue - 1.64292740821838 * has\_underparts\_color\_brown + 1.66115808486938 * has\_underparts\_color\_grey * (-1.14893710613251 * has\_underparts\_color\_grey - 0.398883730173111) - 1.63640189170837 * has\_underparts\_color\_purple - 1.66675746440887 * has\_underparts\_color\_red + 1.53731632232666 * has\_underparts\_color\_rufous + 0.8258376121521 * has\_underparts\_color\_white - 2.24085354804993 * has\_upper\_tail\_color\_green - 2.45426249504089 * has\_upper\_tail\_color\_olive - 2.42568469047546 * has\_upper\_tail\_color\_pink - 3.17689180374146 * has\_upperparts\_color\_buff - 1.25888073444366 * has\_upperparts\_color\_red + 1.48992025852203 * (-1.14381647109985 * has\_underparts\_color\_iridescent + (has\_upper\_tail\_color\_olive - 1.51177990436554) * (has\_upper\_tail\_color\_olive + 0.373325765132904)) * (-0.848836362361908 * has\_underparts\_color\_blue + has\_underparts\_color\_rufous + 0.892222940921783) + 1.045170545578$ |

*Table 8.* Predictive performance and model complexity for all methods on the AWA2 and AWA2-Incomplete datasets.

| Dataset | Model | Accuracy | Nodes | Depth | Expr.-Comp. | Vars | Ops | Weighted | Pareto |
|---|---|---|---|---|---|---|---|---|---|
| AWA2 | BlackBox | $97.6_{\pm0.1}$ | – | – | – | – | – | – | 0/6 |
| | CEM | $97.6_{\pm0.1}$ | – | – | – | – | – | – | 0/6 |
| | LICEM | $97.6_{\pm0.2}$ | – | – | – | – | – | – | 0/6 |
| | DCR | $26.0_{\pm13.1}$ | – | – | – | – | – | – | 0/6 |
| | CMR (1) | $95.3_{\pm1.4}$ | $127.0_{\pm95.6}$ | $5.4_{\pm3.9}$ | $299.8_{\pm224.8}$ | $77.6_{\pm59.4}$ | $49.4_{\pm36.2}$ | $176.4_{\pm131.8}$ | 6/6 |
| | CMR (2) | $96.4_{\pm0.4}$ | $918.4_{\pm1271.6}$ | $22.8_{\pm30.9}$ | $2172.8_{\pm2994.5}$ | $567.2_{\pm799.6}$ | $351.2_{\pm472.0}$ | $1269.6_{\pm1743.6}$ | 0/6 |
| | CMR (3) | $96.6_{\pm0.4}$ | $1400.6_{\pm986.3}$ | $35.4_{\pm26.0}$ | $3324.2_{\pm2332.5}$ | $854.0_{\pm609.1}$ | $546.6_{\pm377.4}$ | $1947.2_{\pm1363.5}$ | 6/6 |
| | CMR (4) | $96.3_{\pm0.5}$ | $928.6_{\pm1416.4}$ | $24.0_{\pm34.1}$ | $2204.6_{\pm3363.3}$ | $565.2_{\pm863.1}$ | $363.4_{\pm553.3}$ | $1292.0_{\pm1969.7}$ | 0/6 |
| | CMR (5) | $96.5_{\pm0.5}$ | $889.6_{\pm1095.7}$ | $22.8_{\pm26.6}$ | $2115.2_{\pm2608.3}$ | $538.4_{\pm661.2}$ | $351.2_{\pm434.6}$ | $1240.8_{\pm1530.3}$ | 6/6 |
| | MLP-M-CBE (1) | $97.5_{\pm0.1}$ | $1105100.0_{\pm0.0}$ | $300.0_{\pm0.0}$ | $6213650.0_{\pm0.0}$ | $4250.0_{\pm0.0}$ | $374050.0_{\pm0.0}$ | $1109350.0_{\pm0.0}$ | 0/6 |
| | MLP-M-CBE (2) | $97.7_{\pm0.1}$ | $1379164.8_{\pm12105.8}$ | $374.4_{\pm3.3}$ | $7754635.2_{\pm68067.1}$ | $5304.0_{\pm46.6}$ | $466814.4_{\pm4097.5}$ | $1384468.8_{\pm12152.3}$ | 6/6 |
| | MLP-M-CBE (3) | $97.6_{\pm0.1}$ | $1410084.8_{\pm65189.1}$ | $382.8_{\pm17.7}$ | $7928488.2_{\pm366538.9}$ | $5423.0_{\pm250.7}$ | $477280.2_{\pm22065.0}$ | $1415507.8_{\pm65439.8}$ | 0/6 |
| | MLP-M-CBE (4) | $97.6_{\pm0.1}$ | $1445455.2_{\pm89237.5}$ | $392.4_{\pm24.2}$ | $8127365.8_{\pm501756.3}$ | $5559.0_{\pm343.2}$ | $489252.2_{\pm30204.7}$ | $1451014.2_{\pm89580.7}$ | 0/6 |
| | MLP-M-CBE (5) | $97.6_{\pm0.1}$ | $1423356.8_{\pm40166.0}$ | $386.4_{\pm10.9}$ | $8003113.2_{\pm225842.0}$ | $5474.0_{\pm154.4}$ | $481772.4_{\pm13595.1}$ | $1428830.8_{\pm40320.4}$ | 0/6 |
| | Lin-M-CBE (1) | $97.6_{\pm0.1}$ | $12847.0_{\pm2.1}$ | $300.0_{\pm0.0}$ | $34142.0_{\pm5.7}$ | $4249.0_{\pm0.7}$ | $4299.0_{\pm0.7}$ | $12847.0_{\pm2.1}$ | 6/6 |
| | Lin-M-CBE (2) | $97.6_{\pm0.1}$ | $14081.8_{\pm216.3}$ | $164.4_{\pm2.5}$ | $37423.6_{\pm574.4}$ | $4657.4_{\pm71.5}$ | $4712.2_{\pm72.4}$ | $14081.8_{\pm216.3}$ | 0/6 |
| | Lin-M-CBE (3) | $97.6_{\pm0.1}$ | $14234.2_{\pm825.8}$ | $166.2_{\pm9.6}$ | $37828.6_{\pm2194.8}$ | $4707.8_{\pm273.1}$ | $4763.2_{\pm276.3}$ | $14234.2_{\pm825.8}$ | 0/6 |
| | Lin-M-CBE (4) | $97.6_{\pm0.1}$ | $13870.8_{\pm316.6}$ | $162.0_{\pm3.7}$ | $36862.8_{\pm841.4}$ | $4587.6_{\pm104.7}$ | $4641.6_{\pm105.9}$ | $13870.8_{\pm316.6}$ | 6/6 |
| | Lin-M-CBE (5) | $97.5_{\pm0.1}$ | $14437.4_{\pm709.4}$ | $168.6_{\pm8.3}$ | $38368.6_{\pm1885.2}$ | $4775.0_{\pm234.6}$ | $4831.2_{\pm237.4}$ | $14437.4_{\pm709.4}$ | 0/6 |
| | Sym-M-CBE (1) | $97.0_{\pm0.1}$ | $15838.7_{\pm226.2}$ | $671.0_{\pm27.0}$ | $107720.0_{\pm5885.9}$ | $2112.0_{\pm56.5}$ | $5636.7_{\pm56.5}$ | $15849.7_{\pm230.8}$ | 0/6 |
| | Sym-M-CBE (2) | $97.0_{\pm0.0}$ | $9419.7_{\pm853.3}$ | $556.7_{\pm54.4}$ | $64022.3_{\pm8817.1}$ | $1473.7_{\pm105.6}$ | $3325.0_{\pm327.6}$ | $9428.0_{\pm854.0}$ | 0/6 |
| | Sym-M-CBE (3) | $97.1_{\pm0.1}$ | $9551.0_{\pm528.1}$ | $604.3_{\pm29.6}$ | $66481.0_{\pm1260.7}$ | $1464.7_{\pm62.7}$ | $3380.3_{\pm177.2}$ | $9557.3_{\pm531.1}$ | 0/6 |
| | Sym-M-CBE (4) | $97.1_{\pm0.1}$ | $8264.7_{\pm1074.0}$ | $538.7_{\pm63.1}$ | $53462.0_{\pm10607.0}$ | $1309.7_{\pm147.5}$ | $2908.3_{\pm372.6}$ | $8272.0_{\pm1077.0}$ | 4/6 |
| | Sym-M-CBE (5) | $97.2_{\pm0.1}$ | $9899.3_{\pm600.6}$ | $633.7_{\pm26.3}$ | $69109.0_{\pm5813.6}$ | $1547.0_{\pm53.8}$ | $3515.3_{\pm230.4}$ | $9907.7_{\pm597.7}$ | 4/6 |
| AWA2-Incomplete | BlackBox | $97.6_{\pm0.1}$ | – | – | – | – | – | – | 0/6 |
| | CEM | $97.4_{\pm0.1}$ | – | – | – | – | – | – | 0/6 |
| | LICEM | $97.5_{\pm0.1}$ | – | – | – | – | – | – | 0/6 |
| | DCR | $96.6_{\pm0.0}$ | – | – | – | – | – | – | 0/6 |
| | CMR (1) | $74.1_{\pm0.4}$ | $615.4_{\pm12.2}$ | $121.2_{\pm2.7}$ | $1401.8_{\pm26.9}$ | $363.6_{\pm8.0}$ | $251.8_{\pm4.4}$ | $867.2_{\pm16.5}$ | 0/6 |
| | CMR (2) | $96.5_{\pm0.1}$ | $591.2_{\pm118.0}$ | $119.2_{\pm26.3}$ | $1336.2_{\pm258.7}$ | $357.8_{\pm78.5}$ | $233.4_{\pm40.8}$ | $824.6_{\pm158.1}$ | 0/6 |
| | CMR (3) | $96.6_{\pm0.1}$ | $535.4_{\pm135.8}$ | $107.4_{\pm29.0}$ | $1212.8_{\pm301.3}$ | $321.8_{\pm87.3}$ | $213.6_{\pm49.2}$ | $749.0_{\pm184.7}$ | 0/6 |
| | CMR (4) | $96.6_{\pm0.1}$ | $442.8_{\pm98.1}$ | $87.0_{\pm20.9}$ | $1011.0_{\pm216.9}$ | $259.4_{\pm63.7}$ | $183.4_{\pm34.5}$ | $626.2_{\pm132.6}$ | 5/6 |
| | CMR (5) | $96.6_{\pm0.1}$ | $451.0_{\pm101.2}$ | $90.0_{\pm24.1}$ | $1023.8_{\pm216.5}$ | $269.2_{\pm71.1}$ | $181.8_{\pm30.3}$ | $632.8_{\pm131.4}$ | 6/6 |
| | MLP-M-CBE (1) | $74.5_{\pm0.1}$ | $10788.0_{\pm242.6}$ | $223.2_{\pm5.0}$ | $57027.6_{\pm1282.6}$ | $334.8_{\pm7.5}$ | $4054.8_{\pm91.2}$ | $11122.8_{\pm250.2}$ | 0/6 |
| | MLP-M-CBE (2) | $96.7_{\pm0.3}$ | $18908.0_{\pm378.1}$ | $391.2_{\pm7.8}$ | $99951.6_{\pm1998.8}$ | $586.8_{\pm11.7}$ | $7106.8_{\pm142.1}$ | $19494.8_{\pm389.8}$ | 0/6 |
| | MLP-M-CBE (3) | $97.0_{\pm0.2}$ | $18328.0_{\pm1270.7}$ | $379.2_{\pm26.3}$ | $96885.6_{\pm6717.3}$ | $568.8_{\pm39.4}$ | $6888.8_{\pm477.6}$ | $18896.8_{\pm1310.2}$ | 1/6 |
| | MLP-M-CBE (4) | $97.1_{\pm0.2}$ | $19488.0_{\pm1202.7}$ | $403.2_{\pm24.9}$ | $103017.6_{\pm6357.8}$ | $604.8_{\pm37.3}$ | $7324.8_{\pm452.1}$ | $20092.8_{\pm1240.0}$ | 0/6 |
| | MLP-M-CBE (5) | $97.1_{\pm0.1}$ | $19082.0_{\pm1053.6}$ | $394.8_{\pm21.8}$ | $100871.4_{\pm5569.7}$ | $592.2_{\pm32.7}$ | $7172.2_{\pm396.0}$ | $19674.2_{\pm1086.3}$ | 0/6 |
| | Lin-M-CBE (1) | $74.2_{\pm0.3}$ | $1119.4_{\pm33.1}$ | $115.8_{\pm3.4}$ | $2895.0_{\pm85.5}$ | $347.4_{\pm10.3}$ | $386.0_{\pm11.4}$ | $1119.4_{\pm33.1}$ | 0/6 |
| | Lin-M-CBE (2) | $97.3_{\pm0.1}$ | $1925.6_{\pm60.1}$ | $199.2_{\pm6.2}$ | $4980.0_{\pm155.5}$ | $597.6_{\pm18.7}$ | $664.0_{\pm20.7}$ | $1925.6_{\pm60.1}$ | 6/6 |
| | Lin-M-CBE (3) | $97.3_{\pm0.1}$ | $1983.6_{\pm105.8}$ | $205.2_{\pm10.9}$ | $5130.0_{\pm273.5}$ | $615.6_{\pm32.8}$ | $684.0_{\pm36.5}$ | $1983.6_{\pm105.8}$ | 6/6 |
| | Lin-M-CBE (4) | $97.3_{\pm0.1}$ | $1849.0_{\pm84.0}$ | $191.4_{\pm8.6}$ | $4781.8_{\pm217.3}$ | $573.8_{\pm26.1}$ | $637.6_{\pm29.0}$ | $1849.0_{\pm84.0}$ | 6/6 |
| | Lin-M-CBE (5) | $97.3_{\pm0.1}$ | $1977.8_{\pm75.1}$ | $204.6_{\pm7.8}$ | $5115.0_{\pm194.1}$ | $613.8_{\pm23.3}$ | $682.0_{\pm25.9}$ | $1977.8_{\pm75.1}$ | 0/6 |
| | Sym-M-CBE (1) | $74.4_{\pm0.1}$ | $1106.3_{\pm24.9}$ | $121.3_{\pm4.9}$ | $2908.0_{\pm75.4}$ | $333.0_{\pm9.0}$ | $383.7_{\pm9.0}$ | $1110.7_{\pm24.8}$ | 0/6 |
| | Sym-M-CBE (2) | $96.5_{\pm0.9}$ | $1097.3_{\pm54.2}$ | $226.7_{\pm6.4}$ | $3290.3_{\pm170.3}$ | $309.7_{\pm17.6}$ | $392.3_{\pm21.5}$ | $1101.7_{\pm51.8}$ | 0/6 |
| | Sym-M-CBE (3) | $96.8_{\pm0.3}$ | $1257.3_{\pm132.3}$ | $260.7_{\pm22.4}$ | $3823.7_{\pm351.4}$ | $363.7_{\pm45.4}$ | $443.0_{\pm43.6}$ | $1262.3_{\pm134.3}$ | 5/6 |
| | Sym-M-CBE (4) | $96.9_{\pm0.1}$ | $1522.0_{\pm94.7}$ | $305.3_{\pm15.9}$ | $4724.3_{\pm306.0}$ | $419.7_{\pm28.5}$ | $547.3_{\pm37.2}$ | $1527.7_{\pm96.5}$ | 0/6 |
| | Sym-M-CBE (5) | $96.9_{\pm0.1}$ | $1469.7_{\pm170.4}$ | $291.0_{\pm18.2}$ | $4390.7_{\pm430.6}$ | $411.0_{\pm52.8}$ | $518.3_{\pm53.4}$ | $1477.7_{\pm171.4}$ | 5/6 |

## M.2. Concept accuracy

In this subsection, we report the concept prediction performance for all methods across the different datasets (Table 13). Specifically, we show the concept accuracy for datasets having binary concepts, and MAE and MSE for datasets having continuous concepts.

*Table 9.* Predictive performance and model complexity for all methods on the CUB200 and CUB200-Incomplete datasets.

| Dataset | Model | Accuracy | Nodes | Depth | Expr.-Comp. | Vars | Ops | Weighted | Pareto |
|---|---|---|---|---|---|---|---|---|---|
| CUB200 | BlackBox | $81.7_{\pm0.3}$ | – | – | – | – | – | – | 0/6 |
| | CEM | $81.8_{\pm0.4}$ | – | – | – | – | – | – | 0/6 |
| | LICEM | $70.4_{\pm2.9}$ | – | – | – | – | – | – | 0/6 |
| | DCR | $19.8_{\pm2.8}$ | – | – | – | – | – | – | 0/6 |
| | CMR (1) | $0.5_{\pm0.3}$ | $556.6_{\pm449.2}$ | $16.2_{\pm12.8}$ | $1300.8_{\pm1045.3}$ | $358.2_{\pm293.8}$ | $198.4_{\pm155.7}$ | $755.0_{\pm604.7}$ | 0/6 |
| | CMR (2) | $0.5_{\pm0.3}$ | $1769.4_{\pm753.4}$ | $51.0_{\pm21.3}$ | $4131.2_{\pm1755.8}$ | $1143.0_{\pm490.4}$ | $626.4_{\pm263.4}$ | $2395.8_{\pm1016.6}$ | 0/6 |
| | CMR (3) | $0.5_{\pm0.5}$ | $1859.4_{\pm1131.8}$ | $53.4_{\pm32.4}$ | $4338.6_{\pm2635.0}$ | $1204.0_{\pm738.7}$ | $655.4_{\pm393.2}$ | $2514.8_{\pm1524.9}$ | 1/6 |
| | CMR (4) | $0.6_{\pm0.4}$ | $1854.6_{\pm708.0}$ | $55.2_{\pm20.4}$ | $4326.4_{\pm1650.8}$ | $1200.6_{\pm459.9}$ | $654.0_{\pm248.7}$ | $2508.6_{\pm956.4}$ | 6/6 |
| | CMR (5) | $0.4_{\pm0.3}$ | $2874.0_{\pm1060.7}$ | $84.0_{\pm30.5}$ | $6706.4_{\pm2465.8}$ | $1859.6_{\pm696.1}$ | $1014.4_{\pm364.9}$ | $3888.4_{\pm1425.5}$ | 0/6 |
| | MLP-M-CBE (1) | $70.4_{\pm0.4}$ | $7638680.0_{\pm268.3}$ | $1200.0_{\pm0.0}$ | $43030320.0_{\pm1520.5}$ | $22400.0_{\pm0.0}$ | $2576160.0_{\pm89.4}$ | $7661080.0_{\pm268.3}$ | 1/6 |
| | MLP-M-CBE (2) | $75.5_{\pm0.6}$ | $9937863.8_{\pm155989.7}$ | $1561.2_{\pm24.5}$ | $55982117.2_{\pm878718.2}$ | $29142.4_{\pm457.7}$ | $3351564.0_{\pm52608.5}$ | $9967005.8_{\pm156447.6}$ | 6/6 |
| | MLP-M-CBE (3) | $75.4_{\pm0.6}$ | $9999189.2_{\pm252485.2}$ | $1570.8_{\pm39.7}$ | $56327579.0_{\pm1422303.1}$ | $29321.6_{\pm740.4}$ | $3372245.8_{\pm85151.1}$ | $10028510.8_{\pm253225.5}$ | 0/6 |
| | MLP-M-CBE (4) | $75.2_{\pm0.5}$ | $9876860.6_{\pm203143.9}$ | $1551.6_{\pm31.9}$ | $55638474.4_{\pm1144353.8}$ | $28963.2_{\pm595.8}$ | $3330990.4_{\pm68510.6}$ | $9905823.6_{\pm203739.6}$ | 6/6 |
| | MLP-M-CBE (5) | $74.7_{\pm0.5}$ | $9838715.0_{\pm163454.9}$ | $1545.6_{\pm25.7}$ | $55423591.4_{\pm920778.8}$ | $28851.2_{\pm479.1}$ | $3318125.8_{\pm55125.2}$ | $9867566.2_{\pm163934.0}$ | 6/6 |
| | Lin-M-CBE (1) | $68.6_{\pm1.1}$ | $67458.2_{\pm183.2}$ | $598.8_{\pm1.6}$ | $179422.8_{\pm487.3}$ | $22353.0_{\pm60.7}$ | $22552.6_{\pm61.3}$ | $67458.2_{\pm183.2}$ | 2/6 |
| | Lin-M-CBE (2) | $70.0_{\pm0.8}$ | $81508.2_{\pm2394.1}$ | $723.6_{\pm21.3}$ | $216792.4_{\pm6367.8}$ | $27008.6_{\pm793.3}$ | $27249.8_{\pm800.4}$ | $81508.2_{\pm2394.1}$ | 5/6 |
| | Lin-M-CBE (3) | $70.0_{\pm0.8}$ | $84550.8_{\pm3304.1}$ | $750.6_{\pm29.3}$ | $224885.0_{\pm8788.2}$ | $28016.8_{\pm1094.9}$ | $28267.0_{\pm1104.6}$ | $84550.8_{\pm3304.1}$ | 0/6 |
| | Lin-M-CBE (4) | $70.4_{\pm0.6}$ | $88460.2_{\pm4598.3}$ | $785.4_{\pm40.9}$ | $235283.0_{\pm12230.2}$ | $29312.2_{\pm1523.7}$ | $29574.0_{\pm1537.3}$ | $88460.2_{\pm4598.3}$ | 5/6 |
| | Lin-M-CBE (5) | $70.5_{\pm0.9}$ | $89130.0_{\pm2078.7}$ | $791.4_{\pm18.4}$ | $237064.6_{\pm5529.1}$ | $29534.2_{\pm688.9}$ | $29798.0_{\pm695.1}$ | $89130.0_{\pm2078.7}$ | 5/6 |
| | Sym-M-CBE (1) | $63.9_{\pm0.4}$ | $65408.0_{\pm459.0}$ | $2190.0_{\pm44.0}$ | $373305.0_{\pm4470.0}$ | $10956.0_{\pm330.0}$ | $22598.0_{\pm570.0}$ | $65431.0_{\pm460.0}$ | 0/6 |
| | Sym-M-CBE (2) | $69.0_{\pm0.7}$ | $60210.0_{\pm420.0}$ | $2667.0_{\pm53.0}$ | $399542.0_{\pm4000.0}$ | $9656.0_{\pm290.0}$ | $20979.0_{\pm620.0}$ | $60234.0_{\pm420.0}$ | 3/6 |
| | Sym-M-CBE (3) | $69.7_{\pm0.7}$ | $60680.0_{\pm430.0}$ | $2809.0_{\pm56.0}$ | $425458.0_{\pm4300.0}$ | $9622.0_{\pm290.0}$ | $21149.0_{\pm630.0}$ | $60713.0_{\pm430.0}$ | 4/6 |
| | Sym-M-CBE (4) | $69.4_{\pm0.7}$ | $73574.0_{\pm520.0}$ | $2987.0_{\pm60.0}$ | $478115.0_{\pm4800.0}$ | $11806.0_{\pm590.0}$ | $25596.0_{\pm640.0}$ | $73614.0_{\pm520.0}$ | 0/6 |
| | Sym-M-CBE (5) | $69.7_{\pm0.7}$ | $71656.0_{\pm510.0}$ | $3185.0_{\pm64.0}$ | $481466.0_{\pm4850.0}$ | $11364.0_{\pm570.0}$ | $24949.0_{\pm625.0}$ | $71689.0_{\pm510.0}$ | 0/6 |
| CUB200-Incomplete | BlackBox | $81.3_{\pm1.2}$ | – | – | – | – | – | – | 0/6 |
| | CEM | $77.7_{\pm1.8}$ | – | – | – | – | – | – | 0/6 |
| | LICEM | $73.0_{\pm2.8}$ | – | – | – | – | – | – | 0/6 |
| | DCR | $61.5_{\pm0.9}$ | – | – | – | – | – | – | 0/6 |
| | CMR (1) | $53.5_{\pm0.8}$ | $6269.2_{\pm236.1}$ | $468.0_{\pm18.1}$ | $15063.6_{\pm563.3}$ | $3432.0_{\pm132.9}$ | $2837.2_{\pm103.3}$ | $9106.4_{\pm339.3}$ | 0/6 |
| | CMR (2) | $61.0_{\pm0.8}$ | $5552.0_{\pm359.9}$ | $496.8_{\pm29.1}$ | $13039.8_{\pm882.4}$ | $3284.2_{\pm201.7}$ | $2267.8_{\pm187.7}$ | $7819.8_{\pm537.4}$ | 6/6 |
| | CMR (3) | $61.0_{\pm0.8}$ | $5128.4_{\pm970.8}$ | $456.8_{\pm89.9}$ | $12058.8_{\pm2269.1}$ | $3021.0_{\pm591.3}$ | $2107.2_{\pm391.3}$ | $7235.6_{\pm1357.0}$ | 6/6 |
| | CMR (4) | $60.8_{\pm0.6}$ | $4301.4_{\pm726.6}$ | $375.6_{\pm59.0}$ | $10155.6_{\pm1743.6}$ | $2498.2_{\pm410.0}$ | $1803.2_{\pm333.1}$ | $6104.6_{\pm1053.4}$ | 6/6 |
| | CMR (5) | $60.6_{\pm0.9}$ | $3946.0_{\pm579.3}$ | $345.8_{\pm72.7}$ | $9317.8_{\pm1318.0}$ | $2289.4_{\pm403.9}$ | $1656.6_{\pm227.5}$ | $5602.6_{\pm782.0}$ | 6/6 |
| | MLP-M-CBE (1) | $56.1_{\pm0.2}$ | $295908.8_{\pm2319.8}$ | $1135.2_{\pm8.9}$ | $1628066.0_{\pm12763.3}$ | $4162.4_{\pm32.6}$ | $104249.2_{\pm817.3}$ | $300071.2_{\pm2352.4}$ | 0/6 |
| | MLP-M-CBE (2) | $75.4_{\pm0.2}$ | $466370.6_{\pm8429.1}$ | $1789.2_{\pm32.4}$ | $2565932.8_{\pm46376.0}$ | $6560.4_{\pm118.9}$ | $164303.2_{\pm2969.6}$ | $472930.8_{\pm8547.7}$ | 6/6 |
| | MLP-M-CBE (3) | $76.2_{\pm0.0}$ | $507674.4_{\pm5810.0}$ | $1947.6_{\pm22.3}$ | $2793183.0_{\pm31966.2}$ | $7141.2_{\pm81.7}$ | $178854.6_{\pm2046.9}$ | $514815.6_{\pm5891.7}$ | 6/6 |
| | MLP-M-CBE (4) | $76.5_{\pm0.8}$ | $510162.6_{\pm6735.4}$ | $1957.2_{\pm25.9}$ | $2806872.8_{\pm37057.4}$ | $7176.4_{\pm95.1}$ | $179731.2_{\pm2372.9}$ | $517338.8_{\pm6830.1}$ | 6/6 |
| | MLP-M-CBE (5) | $77.2_{\pm0.7}$ | $519248.0_{\pm14629.9}$ | $1992.0_{\pm56.1}$ | $2856860.0_{\pm80492.4}$ | $7304.0_{\pm205.8}$ | $182932.0_{\pm5154.1}$ | $526552.0_{\pm14835.7}$ | 6/6 |
| | Lin-M-CBE (1) | $54.6_{\pm0.9}$ | $12539.2_{\pm156.5}$ | $553.2_{\pm6.9}$ | $33007.6_{\pm412.1}$ | $4056.8_{\pm50.6}$ | $4241.2_{\pm52.9}$ | $12539.2_{\pm156.5}$ | 0/6 |
| | Lin-M-CBE (2) | $68.4_{\pm1.5}$ | $18413.2_{\pm373.8}$ | $812.4_{\pm16.5}$ | $48470.0_{\pm983.9}$ | $5957.2_{\pm120.9}$ | $6228.0_{\pm126.4}$ | $18413.2_{\pm373.8}$ | 2/6 |
| | Lin-M-CBE (3) | $68.5_{\pm1.0}$ | $18658.0_{\pm567.1}$ | $823.2_{\pm25.0}$ | $49114.4_{\pm1493.0}$ | $6036.4_{\pm183.5}$ | $6310.8_{\pm191.8}$ | $18658.0_{\pm567.1}$ | 2/6 |
| | Lin-M-CBE (4) | $68.6_{\pm0.5}$ | $19061.2_{\pm314.3}$ | $841.2_{\pm13.8}$ | $50175.6_{\pm827.4}$ | $6166.8_{\pm101.7}$ | $6447.2_{\pm106.3}$ | $19061.2_{\pm314.3}$ | 2/6 |
| | Lin-M-CBE (5) | $69.1_{\pm0.6}$ | $19446.8_{\pm675.6}$ | $858.0_{\pm29.8}$ | $51190.8_{\pm1778.3}$ | $6291.6_{\pm218.6}$ | $6577.6_{\pm228.5}$ | $19446.8_{\pm675.6}$ | 2/6 |
| | Sym-M-CBE (1) | $55.6_{\pm1.2}$ | $13182.0_{\pm205.0}$ | $683.0_{\pm32.0}$ | $35933.0_{\pm850.0}$ | $4066.0_{\pm50.0}$ | $4476.0_{\pm120.0}$ | $13188.0_{\pm205.0}$ | 0/6 |
| | Sym-M-CBE (2) | $60.4_{\pm1.5}$ | $16102.0_{\pm310.0}$ | $1359.0_{\pm65.0}$ | $58572.0_{\pm1500.0}$ | $4091.0_{\pm70.0}$ | $5653.0_{\pm150.0}$ | $16126.0_{\pm310.0}$ | 0/6 |
| | Sym-M-CBE (3) | $64.7_{\pm3.9}$ | $15807.8_{\pm1117.5}$ | $1423.0_{\pm107.5}$ | $57834.5_{\pm6948.5}$ | $4039.5_{\pm182.1}$ | $5557.0_{\pm476.5}$ | $15826.5_{\pm1118.5}$ | 2/6 |
| | Sym-M-CBE (4) | $68.9_{\pm6.2}$ | $15513.5_{\pm2124.9}$ | $1487.0_{\pm149.9}$ | $57097.0_{\pm12384.3}$ | $3988.0_{\pm294.2}$ | $5461.0_{\pm803.3}$ | $15527.0_{\pm2127.0}$ | 3/6 |
| | Sym-M-CBE (5) | $70.9_{\pm1.8}$ | $15629.0_{\pm420.0}$ | $1514.0_{\pm80.0}$ | $61364.0_{\pm2100.0}$ | $3884.0_{\pm90.0}$ | $5511.0_{\pm200.0}$ | $15636.0_{\pm420.0}$ | 6/6 |

*Table 10.* Predictive performance and model complexity for all methods on the CIFAR10 dataset.

| Dataset | Model | Accuracy | Nodes | Depth | Expr.-Comp. | Vars | Ops | Weighted | Pareto |
|---|---|---|---|---|---|---|---|---|---|
| CIFAR10 | BlackBox | $87.6_{\pm0.4}$ | – | – | – | – | – | – | 0/6 |
| | CEM | $87.7_{\pm0.1}$ | – | – | – | – | – | – | 0/6 |
| | LICEM | $87.3_{\pm0.2}$ | – | – | – | – | – | – | 0/6 |
| | DCR | $17.2_{\pm16.0}$ | – | – | – | – | – | – | 0/6 |
| | CMR (1) | $10.3_{\pm5.4}$ | $322.0_{\pm206.6}$ | $7.2_{\pm4.5}$ | $749.2_{\pm478.2}$ | $212.0_{\pm138.7}$ | $110.0_{\pm68.1}$ | $432.0_{\pm274.6}$ | 6/6 |
| | CMR (2) | $9.8_{\pm3.4}$ | $399.4_{\pm168.2}$ | $9.0_{\pm3.7}$ | $926.8_{\pm389.6}$ | $265.4_{\pm112.8}$ | $134.0_{\pm55.6}$ | $533.4_{\pm223.8}$ | 0/6 |
| | CMR (3) | $8.9_{\pm4.1}$ | $611.4_{\pm194.6}$ | $13.8_{\pm4.0}$ | $1418.2_{\pm458.5}$ | $406.8_{\pm123.1}$ | $204.6_{\pm72.4}$ | $816.0_{\pm266.6}$ | 0/6 |
| | CMR (4) | $10.3_{\pm2.6}$ | $463.0_{\pm249.4}$ | $10.2_{\pm5.4}$ | $1080.2_{\pm580.7}$ | $302.0_{\pm164.0}$ | $161.0_{\pm85.8}$ | $624.0_{\pm335.0}$ | 5/6 |
| | CMR (5) | $13.6_{\pm5.0}$ | $533.6_{\pm372.6}$ | $12.0_{\pm7.9}$ | $1241.6_{\pm869.0}$ | $351.2_{\pm243.6}$ | $182.4_{\pm129.1}$ | $716.0_{\pm501.7}$ | 5/6 |
| | MLP-M-CBE (1) | $81.8_{\pm0.5}$ | $620634.0_{\pm13.4}$ | $60.0_{\pm0.0}$ | $3500636.0_{\pm76.0}$ | $1430.0_{\pm0.0}$ | $208788.0_{\pm4.5}$ | $622064.0_{\pm13.4}$ | 0/6 |
| | MLP-M-CBE (2) | $85.8_{\pm0.2}$ | $1141966.2_{\pm33990.6}$ | $110.4_{\pm3.3}$ | $6441168.2_{\pm191721.0}$ | $2631.2_{\pm78.3}$ | $384169.8_{\pm11434.8}$ | $1144597.4_{\pm34068.9}$ | 6/6 |
| | MLP-M-CBE (3) | $84.8_{\pm1.9}$ | $1117132.8_{\pm294380.7}$ | $108.0_{\pm28.5}$ | $6301097.2_{\pm1660430.3}$ | $2574.0_{\pm678.3}$ | $375815.6_{\pm99032.9}$ | $1119706.8_{\pm295059.0}$ | 0/6 |
| | MLP-M-CBE (4) | $85.5_{\pm0.4}$ | $1390197.6_{\pm120968.5}$ | $134.4_{\pm11.7}$ | $7841296.8_{\pm682312.5}$ | $3203.2_{\pm278.8}$ | $467677.6_{\pm40695.1}$ | $1393400.8_{\pm121247.2}$ | 0/6 |
| | MLP-M-CBE (5) | $84.3_{\pm1.2}$ | $1166793.6_{\pm278935.6}$ | $112.8_{\pm27.0}$ | $6581205.2_{\pm1573313.7}$ | $2688.4_{\pm642.7}$ | $392522.0_{\pm93837.0}$ | $1169482.0_{\pm279578.3}$ | 0/6 |
| | Lin-M-CBE (1) | $81.5_{\pm0.3}$ | $4308.2_{\pm2.7}$ | $30.0_{\pm0.0}$ | $11465.2_{\pm7.2}$ | $1429.4_{\pm0.9}$ | $1439.4_{\pm0.9}$ | $4308.2_{\pm2.7}$ | 2/6 |
| | Lin-M-CBE (2) | $85.5_{\pm0.3}$ | $8445.8_{\pm234.4}$ | $58.8_{\pm1.6}$ | $22476.4_{\pm623.9}$ | $2802.2_{\pm77.8}$ | $2821.8_{\pm78.3}$ | $8445.8_{\pm234.4}$ | 5/6 |
| | Lin-M-CBE (3) | $85.7_{\pm0.2}$ | $10513.4_{\pm784.6}$ | $73.2_{\pm5.4}$ | $27978.8_{\pm2088.1}$ | $3488.2_{\pm260.3}$ | $3512.6_{\pm262.1}$ | $10513.4_{\pm784.6}$ | 5/6 |
| | Lin-M-CBE (4) | $85.6_{\pm0.5}$ | $10941.4_{\pm1082.1}$ | $76.2_{\pm7.5}$ | $29117.8_{\pm2879.8}$ | $3630.2_{\pm359.0}$ | $3655.6_{\pm361.5}$ | $10941.4_{\pm1082.1}$ | 0/6 |
| | Lin-M-CBE (5) | $85.6_{\pm0.5}$ | $11290.4_{\pm933.9}$ | $78.6_{\pm6.5}$ | $30046.6_{\pm2485.3}$ | $3746.0_{\pm309.9}$ | $3772.2_{\pm312.0}$ | $11290.4_{\pm933.9}$ | 0/6 |
| | Sym-M-CBE (1) | $78.6_{\pm0.3}$ | $1058.0_{\pm147.2}$ | $56.3_{\pm6.1}$ | $3639.3_{\pm837.8}$ | $299.3_{\pm35.7}$ | $352.7_{\pm51.2}$ | $1058.0_{\pm147.2}$ | 5/6 |
| | Sym-M-CBE (2) | $84.5_{\pm1.1}$ | $3733.7_{\pm810.1}$ | $149.0_{\pm24.3}$ | $18092.7_{\pm5546.4}$ | $737.3_{\pm131.5}$ | $1279.3_{\pm269.3}$ | $3736.3_{\pm812.1}$ | 5/6 |
| | Sym-M-CBE (3) | $84.9_{\pm0.4}$ | $5197.7_{\pm882.0}$ | $195.3_{\pm32.7}$ | $29891.3_{\pm4271.5}$ | $895.3_{\pm101.5}$ | $1784.7_{\pm301.9}$ | $5198.7_{\pm882.0}$ | 4/6 |
| | Sym-M-CBE (4) | $85.0_{\pm0.9}$ | $6988.3_{\pm1143.1}$ | $244.0_{\pm28.6}$ | $38697.7_{\pm6142.7}$ | $1184.3_{\pm221.5}$ | $2407.0_{\pm397.1}$ | $6990.3_{\pm1141.4}$ | 0/6 |
| | Sym-M-CBE (5) | $85.3_{\pm0.2}$ | $6540.0_{\pm327.2}$ | $258.0_{\pm34.1}$ | $47484.7_{\pm8442.5}$ | $1057.3_{\pm51.4}$ | $2270.0_{\pm118.5}$ | $6544.0_{\pm325.9}$ | 4/6 |

*Table 11.* Predictive performance and model complexity for all methods on the dSprites-Exp. and Pendulum datasets.

| Dataset | Model | MAE | MSE | Nodes | Depth | Expr.-Comp. | Vars | Ops | Weighted | Pareto |
|---|---|---|---|---|---|---|---|---|---|---|
| dSprites-Exp. | BlackBox | $0.933_{\pm0.033}$ | $1.778_{\pm0.060}$ | – | – | – | – | – | – | 0/6 |
| | CEM | $0.880_{\pm0.028}$ | $1.972_{\pm0.137}$ | – | – | – | – | – | – | 0/6 |
| | LICEM | $0.908_{\pm0.011}$ | $2.014_{\pm0.062}$ | – | – | – | – | – | – | 0/6 |
| | MLP-M-CBE (1) | $1.131_{\pm0.120}$ | $3.491_{\pm2.340}$ | $32.8_{\pm12.0}$ | $6.0_{\pm0.0}$ | $145.8_{\pm55.9}$ | $2.0_{\pm0.0}$ | $15.0_{\pm5.5}$ | $35.6_{\pm13.1}$ | 0/6 |
| | MLP-M-CBE (2) | $0.993_{\pm0.092}$ | $2.198_{\pm0.111}$ | $60.4_{\pm30.3}$ | $9.6_{\pm3.3}$ | $270.0_{\pm138.1}$ | $3.2_{\pm1.1}$ | $27.6_{\pm13.8}$ | $65.6_{\pm33.0}$ | 0/6 |
| | MLP-M-CBE (3) | $0.999_{\pm0.103}$ | $2.129_{\pm0.219}$ | $92.8_{\pm48.1}$ | $14.4_{\pm5.4}$ | $415.2_{\pm218.5}$ | $4.8_{\pm1.8}$ | $42.4_{\pm21.9}$ | $100.8_{\pm52.3}$ | 0/6 |
| | MLP-M-CBE (4) | $0.959_{\pm0.038}$ | $2.199_{\pm0.246}$ | $93.0_{\pm67.0}$ | $15.0_{\pm7.7}$ | $415.5_{\pm303.7}$ | $5.0_{\pm2.6}$ | $42.5_{\pm30.6}$ | $101.0_{\pm72.9}$ | 0/6 |
| | MLP-M-CBE (5) | $1.004_{\pm0.087}$ | $2.214_{\pm0.255}$ | $128.0_{\pm84.4}$ | $21.0_{\pm11.5}$ | $571.5_{\pm381.7}$ | $7.0_{\pm3.8}$ | $58.5_{\pm38.5}$ | $139.0_{\pm91.8}$ | 0/6 |
| | Prior-M-CBE (1) | $0.902_{\pm0.007}$ | $2.146_{\pm0.027}$ | $10.0_{\pm0.0}$ | $5.0_{\pm0.0}$ | $37.0_{\pm0.0}$ | $2.0_{\pm0.0}$ | $6.0_{\pm0.0}$ | $13.0_{\pm0.0}$ | 6/6 |
| | Lin-M-CBE (1) | $1.128_{\pm0.006}$ | $2.307_{\pm0.028}$ | $7.4_{\pm1.3}$ | $3.0_{\pm0.0}$ | $17.4_{\pm3.6}$ | $1.8_{\pm0.4}$ | $2.8_{\pm0.4}$ | $7.4_{\pm1.3}$ | 6/6 |
| | Lin-M-CBE (2) | $1.018_{\pm0.069}$ | $2.032_{\pm0.170}$ | $13.8_{\pm4.9}$ | $5.4_{\pm1.3}$ | $32.6_{\pm12.1}$ | $3.4_{\pm1.3}$ | $5.2_{\pm1.8}$ | $13.8_{\pm4.9}$ | 2/6 |
| | Lin-M-CBE (3) | $0.967_{\pm0.027}$ | $1.884_{\pm0.055}$ | $24.0_{\pm0.0}$ | $9.0_{\pm0.0}$ | $57.0_{\pm0.0}$ | $6.0_{\pm0.0}$ | $9.0_{\pm0.0}$ | $24.0_{\pm0.0}$ | 0/6 |
| | Lin-M-CBE (4) | $1.022_{\pm0.072}$ | $2.024_{\pm0.211}$ | $26.0_{\pm12.0}$ | $9.8_{\pm4.5}$ | $61.8_{\pm28.5}$ | $6.5_{\pm3.0}$ | $9.8_{\pm4.5}$ | $26.0_{\pm12.0}$ | 0/6 |
| | Lin-M-CBE (5) | $0.944_{\pm0.016}$ | $1.810_{\pm0.021}$ | $37.2_{\pm5.5}$ | $14.2_{\pm1.5}$ | $88.2_{\pm13.5}$ | $9.2_{\pm1.5}$ | $14.0_{\pm2.0}$ | $37.2_{\pm5.5}$ | 0/6 |
| | Sym-M-CBE (1) | $1.014_{\pm0.016}$ | $2.611_{\pm0.092}$ | $16.4_{\pm5.9}$ | $7.4_{\pm2.1}$ | $83.2_{\pm49.8}$ | $2.0_{\pm0.0}$ | $9.4_{\pm3.4}$ | $20.0_{\pm7.1}$ | 0/6 |
| | Sym-M-CBE (2) | $1.011_{\pm0.030}$ | $2.615_{\pm0.137}$ | $13.2_{\pm1.8}$ | $6.6_{\pm0.9}$ | $57.6_{\pm13.2}$ | $2.0_{\pm0.0}$ | $7.6_{\pm0.9}$ | $16.2_{\pm1.8}$ | 0/6 |
| | Sym-M-CBE (3) | $1.009_{\pm0.041}$ | $2.598_{\pm0.204}$ | $15.2_{\pm3.0}$ | $7.6_{\pm1.8}$ | $72.6_{\pm23.5}$ | $2.0_{\pm0.0}$ | $9.0_{\pm2.0}$ | $19.2_{\pm4.4}$ | 0/6 |
| | Sym-M-CBE (4) | $0.991_{\pm0.029}$ | $2.512_{\pm0.158}$ | $12.5_{\pm3.8}$ | $5.8_{\pm1.0}$ | $52.5_{\pm23.7}$ | $2.0_{\pm0.0}$ | $7.2_{\pm1.9}$ | $15.5_{\pm3.8}$ | 0/6 |
| | Sym-M-CBE (5) | $0.985_{\pm0.015}$ | $2.432_{\pm0.070}$ | $13.5_{\pm1.9}$ | $6.5_{\pm0.6}$ | $58.8_{\pm12.1}$ | $2.0_{\pm0.0}$ | $7.8_{\pm1.0}$ | $16.5_{\pm1.9}$ | 0/6 |
| Pendulum | BlackBox | $0.144_{\pm0.059}$ | $0.036_{\pm0.026}$ | – | – | – | – | – | – | 0/6 |
| | CEM | $0.599_{\pm0.117}$ | $0.678_{\pm0.250}$ | – | – | – | – | – | – | 0/6 |
| | LICEM | $0.774_{\pm0.079}$ | $1.030_{\pm0.206}$ | – | – | – | – | – | – | 0/6 |
| | MLP-M-CBE (1) | $2.445_{\pm0.561}$ | $8.150_{\pm3.317}$ | $37.2_{\pm12.0}$ | $6.0_{\pm0.0}$ | $166.2_{\pm55.9}$ | $2.0_{\pm0.0}$ | $17.0_{\pm5.5}$ | $40.4_{\pm13.1}$ | 0/6 |
| | MLP-M-CBE (2) | $1.519_{\pm0.097}$ | $3.392_{\pm0.275}$ | $74.4_{\pm24.1}$ | $12.0_{\pm0.0}$ | $332.4_{\pm111.7}$ | $4.0_{\pm0.0}$ | $34.0_{\pm11.0}$ | $80.8_{\pm26.3}$ | 0/6 |
| | MLP-M-CBE (3) | $1.185_{\pm0.132}$ | $2.249_{\pm0.479}$ | $111.6_{\pm36.1}$ | $18.0_{\pm0.0}$ | $498.6_{\pm167.6}$ | $6.0_{\pm0.0}$ | $51.0_{\pm16.4}$ | $121.2_{\pm39.4}$ | 0/6 |
| | MLP-M-CBE (4) | $0.997_{\pm0.035}$ | $1.628_{\pm0.096}$ | $140.0_{\pm50.8}$ | $24.0_{\pm0.0}$ | $624.0_{\pm235.6}$ | $8.0_{\pm0.0}$ | $64.0_{\pm23.1}$ | $152.0_{\pm55.4}$ | 0/6 |
| | MLP-M-CBE (5) | $0.895_{\pm0.149}$ | $1.336_{\pm0.436}$ | $175.0_{\pm63.5}$ | $30.0_{\pm0.0}$ | $780.0_{\pm294.4}$ | $10.0_{\pm0.0}$ | $80.0_{\pm28.9}$ | $190.0_{\pm69.3}$ | 0/6 |
| | Prior-M-CBE (1) | $0.237_{\pm0.010}$ | $0.147_{\pm0.011}$ | $6.0_{\pm0.0}$ | $4.0_{\pm0.0}$ | $15.0_{\pm0.0}$ | $1.0_{\pm0.0}$ | $3.0_{\pm0.0}$ | $7.0_{\pm0.0}$ | 6/6 |
| | Lin-M-CBE (1) | $3.040_{\pm0.002}$ | $11.909_{\pm0.016}$ | $8.0_{\pm0.0}$ | $3.0_{\pm0.0}$ | $19.0_{\pm0.0}$ | $2.0_{\pm0.0}$ | $3.0_{\pm0.0}$ | $8.0_{\pm0.0}$ | 1/6 |
| | Lin-M-CBE (2) | $1.735_{\pm0.084}$ | $4.288_{\pm0.446}$ | $16.0_{\pm0.0}$ | $6.0_{\pm0.0}$ | $38.0_{\pm0.0}$ | $4.0_{\pm0.0}$ | $6.0_{\pm0.0}$ | $16.0_{\pm0.0}$ | 0/6 |
| | Lin-M-CBE (3) | $1.150_{\pm0.121}$ | $2.046_{\pm0.338}$ | $24.0_{\pm0.0}$ | $9.0_{\pm0.0}$ | $57.0_{\pm0.0}$ | $6.0_{\pm0.0}$ | $9.0_{\pm0.0}$ | $24.0_{\pm0.0}$ | 0/6 |
| | Lin-M-CBE (4) | $0.909_{\pm0.104}$ | $1.339_{\pm0.272}$ | $32.0_{\pm0.0}$ | $12.0_{\pm0.0}$ | $76.0_{\pm0.0}$ | $8.0_{\pm0.0}$ | $12.0_{\pm0.0}$ | $32.0_{\pm0.0}$ | 0/6 |
| | Lin-M-CBE (5) | $0.720_{\pm0.095}$ | $0.897_{\pm0.260}$ | $40.0_{\pm0.0}$ | $15.0_{\pm0.0}$ | $95.0_{\pm0.0}$ | $10.0_{\pm0.0}$ | $15.0_{\pm0.0}$ | $40.0_{\pm0.0}$ | 0/6 |
| | Sym-M-CBE (1) | $0.331_{\pm0.134}$ | $0.357_{\pm0.342}$ | $13.2_{\pm8.0}$ | $6.0_{\pm2.5}$ | $54.0_{\pm46.1}$ | $1.2_{\pm0.4}$ | $7.0_{\pm4.6}$ | $16.0_{\pm10.5}$ | 0/6 |
| | Sym-M-CBE (2) | $0.261_{\pm0.092}$ | $0.199_{\pm0.139}$ | $6.0_{\pm0.0}$ | $4.0_{\pm0.0}$ | $15.0_{\pm0.0}$ | $1.0_{\pm0.0}$ | $3.0_{\pm0.0}$ | $7.0_{\pm0.0}$ | 0/6 |
| | Sym-M-CBE (3) | $0.348_{\pm0.111}$ | $0.384_{\pm0.242}$ | $10.8_{\pm7.6}$ | $4.8_{\pm1.5}$ | $34.2_{\pm33.4}$ | $1.0_{\pm0.0}$ | $5.8_{\pm4.9}$ | $13.2_{\pm10.6}$ | 0/6 |
| | Sym-M-CBE (4) | $0.267_{\pm0.037}$ | $0.178_{\pm0.040}$ | $6.0_{\pm0.0}$ | $4.0_{\pm0.0}$ | $15.0_{\pm0.0}$ | $1.0_{\pm0.0}$ | $3.0_{\pm0.0}$ | $7.0_{\pm0.0}$ | 0/6 |
| | Sym-M-CBE (5) | $0.358_{\pm0.168}$ | $0.382_{\pm0.298}$ | $8.5_{\pm5.0}$ | $5.5_{\pm3.0}$ | $30.5_{\pm31.0}$ | $1.0_{\pm0.0}$ | $5.0_{\pm4.0}$ | $11.0_{\pm8.0}$ | 0/6 |

*Table 12.* Predictive performance and model complexity for all methods on the MNIST-Arith. and MAWPS datasets.

| Dataset | Model | MAE | MSE | Nodes | Depth | Expr.-Comp. | Vars | Ops | Weighted | Pareto |
|---------|-------|-----|-----|-------|-------|-------------|------|-----|----------|--------|
| MNIST-Arith. | BlackBox | $2.123_{\pm0.026}$ | $16.025_{\pm0.626}$ | – | – | – | – | – | – | 0/6 |
| | CEM | $1.805_{\pm0.211}$ | $15.003_{\pm1.930}$ | – | – | – | – | – | – | 0/6 |
| | LICEM | $1.546_{\pm0.128}$ | $12.927_{\pm1.246}$ | – | – | – | – | – | – | 0/6 |
| | MLP-M-CBE (1) | $8.650_{\pm0.214}$ | $354.843_{\pm301.231}$ | $32.8_{\pm12.0}$ | $6.0_{\pm0.0}$ | $145.8_{\pm55.9}$ | $2.0_{\pm0.0}$ | $15.0_{\pm5.5}$ | $35.6_{\pm13.1}$ | 0/6 |
| | MLP-M-CBE (2) | $5.154_{\pm0.199}$ | $104.757_{\pm51.415}$ | $74.4_{\pm24.1}$ | $12.0_{\pm0.0}$ | $332.4_{\pm111.7}$ | $4.0_{\pm0.0}$ | $34.0_{\pm11.0}$ | $80.8_{\pm26.3}$ | 0/6 |
| | MLP-M-CBE (3) | $2.791_{\pm0.117}$ | $29.396_{\pm8.131}$ | $111.6_{\pm36.1}$ | $18.0_{\pm0.0}$ | $498.6_{\pm167.6}$ | $6.0_{\pm0.0}$ | $51.0_{\pm16.4}$ | $121.2_{\pm39.4}$ | 1/6 |
| | MLP-M-CBE (4) | $2.885_{\pm0.183}$ | $41.241_{\pm23.051}$ | $140.0_{\pm50.8}$ | $24.0_{\pm0.0}$ | $624.0_{\pm235.6}$ | $8.0_{\pm0.0}$ | $64.0_{\pm23.1}$ | $152.0_{\pm55.4}$ | 0/6 |
| | MLP-M-CBE (5) | $2.497_{\pm0.171}$ | $26.950_{\pm6.720}$ | $175.0_{\pm63.5}$ | $30.0_{\pm0.0}$ | $780.0_{\pm294.4}$ | $10.0_{\pm0.0}$ | $80.0_{\pm28.9}$ | $190.0_{\pm69.3}$ | 6/6 |
| | Prior-M-CBE (4) | $2.667_{\pm0.130}$ | $782.072_{\pm777.048}$ | $16.0_{\pm0.0}$ | $10.0_{\pm0.0}$ | $32.0_{\pm0.0}$ | $8.0_{\pm0.0}$ | $6.0_{\pm0.0}$ | $17.0_{\pm0.0}$ | 6/6 |
| | Lin-M-CBE (1) | $8.544_{\pm0.096}$ | $168.821_{\pm2.143}$ | $8.0_{\pm0.0}$ | $3.0_{\pm0.0}$ | $19.0_{\pm0.0}$ | $2.0_{\pm0.0}$ | $3.0_{\pm0.0}$ | $8.0_{\pm0.0}$ | 2/6 |
| | Lin-M-CBE (2) | $5.940_{\pm0.058}$ | $69.892_{\pm1.141}$ | $16.0_{\pm0.0}$ | $6.0_{\pm0.0}$ | $38.0_{\pm0.0}$ | $4.0_{\pm0.0}$ | $6.0_{\pm0.0}$ | $16.0_{\pm0.0}$ | 0/6 |
| | Lin-M-CBE (3) | $3.499_{\pm0.074}$ | $34.529_{\pm1.378}$ | $24.0_{\pm0.0}$ | $9.0_{\pm0.0}$ | $57.0_{\pm0.0}$ | $6.0_{\pm0.0}$ | $9.0_{\pm0.0}$ | $24.0_{\pm0.0}$ | 0/6 |
| | Lin-M-CBE (4) | $2.739_{\pm0.066}$ | $23.332_{\pm1.277}$ | $32.0_{\pm0.0}$ | $12.0_{\pm0.0}$ | $76.0_{\pm0.0}$ | $8.0_{\pm0.0}$ | $12.0_{\pm0.0}$ | $32.0_{\pm0.0}$ | 0/6 |
| | Lin-M-CBE (5) | $2.524_{\pm0.050}$ | $19.624_{\pm0.736}$ | $40.0_{\pm0.0}$ | $15.0_{\pm0.0}$ | $95.0_{\pm0.0}$ | $10.0_{\pm0.0}$ | $15.0_{\pm0.0}$ | $40.0_{\pm0.0}$ | 5/6 |
| | Sym-M-CBE (1) | $8.575_{\pm0.092}$ | $173.141_{\pm2.915}$ | $3.0_{\pm0.0}$ | $2.0_{\pm0.0}$ | $5.0_{\pm0.0}$ | $2.0_{\pm0.0}$ | $1.0_{\pm0.0}$ | $3.0_{\pm0.0}$ | 5/6 |
| | Sym-M-CBE (2) | $5.132_{\pm0.183}$ | $53.198_{\pm6.903}$ | $6.0_{\pm0.0}$ | $4.0_{\pm0.0}$ | $10.0_{\pm0.0}$ | $3.0_{\pm0.0}$ | $2.0_{\pm0.0}$ | $6.0_{\pm0.0}$ | 6/6 |
| | Sym-M-CBE (3) | $2.948_{\pm0.147}$ | $24.491_{\pm2.150}$ | $11.0_{\pm0.0}$ | $7.0_{\pm0.0}$ | $21.0_{\pm0.0}$ | $6.0_{\pm0.0}$ | $4.0_{\pm0.0}$ | $11.0_{\pm0.0}$ | 5/6 |
| | Sym-M-CBE (4) | $2.847_{\pm0.108}$ | $63.271_{\pm78.301}$ | $12.2_{\pm2.5}$ | $7.8_{\pm1.5}$ | $23.8_{\pm5.5}$ | $6.5_{\pm1.0}$ | $4.5_{\pm1.0}$ | $12.5_{\pm3.0}$ | 5/6 |
| | Sym-M-CBE (5) | $3.218_{\pm0.576}$ | $7304.442_{\pm8578.163}$ | $16.0_{\pm0.0}$ | $10.0_{\pm0.0}$ | $32.0_{\pm0.0}$ | $8.0_{\pm0.0}$ | $6.0_{\pm0.0}$ | $17.0_{\pm0.0}$ | 0/6 |
| MAWPS | BlackBox | $0.787_{\pm0.044}$ | $1.292_{\pm0.142}$ | – | – | – | – | – | – | 0/6 |
| | CEM | $0.756_{\pm0.017}$ | $1.149_{\pm0.057}$ | – | – | – | – | – | – | 0/6 |
| | LICEM | $0.768_{\pm0.031}$ | $1.195_{\pm0.091}$ | – | – | – | – | – | – | 0/6 |
| | MLP-M-CBE (1) | $4.968_{\pm0.119}$ | $46.966_{\pm1.458}$ | $60.8_{\pm23.0}$ | $6.0_{\pm0.0}$ | $288.6_{\pm111.7}$ | $3.0_{\pm0.0}$ | $26.2_{\pm9.9}$ | $65.0_{\pm24.6}$ | 1/6 |
| | MLP-M-CBE (2) | $3.451_{\pm0.222}$ | $22.871_{\pm2.562}$ | $138.4_{\pm46.0}$ | $12.0_{\pm0.0}$ | $658.8_{\pm223.5}$ | $6.0_{\pm0.0}$ | $59.6_{\pm19.7}$ | $148.0_{\pm49.3}$ | 1/6 |
| | MLP-M-CBE (3) | $2.460_{\pm0.298}$ | $12.382_{\pm3.186}$ | $207.6_{\pm69.0}$ | $18.0_{\pm0.0}$ | $988.2_{\pm335.2}$ | $9.0_{\pm0.0}$ | $89.4_{\pm29.6}$ | $222.0_{\pm73.9}$ | 0/6 |
| | MLP-M-CBE (4) | $2.164_{\pm0.250}$ | $9.330_{\pm1.999}$ | $260.0_{\pm97.0}$ | $24.0_{\pm0.0}$ | $1236.0_{\pm471.1}$ | $12.0_{\pm0.0}$ | $112.0_{\pm41.6}$ | $278.0_{\pm103.9}$ | 0/6 |
| | MLP-M-CBE (5) | $1.925_{\pm0.352}$ | $7.607_{\pm2.785}$ | $325.0_{\pm121.2}$ | $30.0_{\pm0.0}$ | $1545.0_{\pm588.9}$ | $15.0_{\pm0.0}$ | $140.0_{\pm52.0}$ | $347.5_{\pm129.9}$ | 0/6 |
| | Prior-M-CBE (4) | $0.996_{\pm0.010}$ | $2.121_{\pm0.064}$ | $24.0_{\pm0.0}$ | $14.0_{\pm0.0}$ | $60.0_{\pm0.0}$ | $12.0_{\pm0.0}$ | $10.0_{\pm0.0}$ | $24.0_{\pm0.0}$ | 6/6 |
| | Lin-M-CBE (1) | $5.433_{\pm0.090}$ | $58.836_{\pm1.768}$ | $11.0_{\pm0.0}$ | $3.0_{\pm0.0}$ | $27.0_{\pm0.0}$ | $3.0_{\pm0.0}$ | $4.0_{\pm0.0}$ | $11.0_{\pm0.0}$ | 0/6 |
| | Lin-M-CBE (2) | $4.154_{\pm0.040}$ | $30.719_{\pm0.703}$ | $22.0_{\pm0.0}$ | $6.0_{\pm0.0}$ | $54.0_{\pm0.0}$ | $6.0_{\pm0.0}$ | $8.0_{\pm0.0}$ | $22.0_{\pm0.0}$ | 0/6 |
| | Lin-M-CBE (3) | $2.981_{\pm0.045}$ | $17.304_{\pm0.532}$ | $33.0_{\pm0.0}$ | $9.0_{\pm0.0}$ | $81.0_{\pm0.0}$ | $9.0_{\pm0.0}$ | $12.0_{\pm0.0}$ | $33.0_{\pm0.0}$ | 1/6 |
| | Lin-M-CBE (4) | $2.635_{\pm0.155}$ | $13.376_{\pm1.387}$ | $44.0_{\pm0.0}$ | $12.0_{\pm0.0}$ | $108.0_{\pm0.0}$ | $12.0_{\pm0.0}$ | $16.0_{\pm0.0}$ | $44.0_{\pm0.0}$ | 0/6 |
| | Lin-M-CBE (5) | $2.491_{\pm0.278}$ | $12.036_{\pm2.217}$ | $55.0_{\pm0.0}$ | $15.0_{\pm0.0}$ | $135.0_{\pm0.0}$ | $15.0_{\pm0.0}$ | $20.0_{\pm0.0}$ | $55.0_{\pm0.0}$ | 0/6 |
| | Sym-M-CBE (1) | $5.123_{\pm0.202}$ | $51.517_{\pm3.897}$ | $4.4_{\pm0.9}$ | $3.0_{\pm0.0}$ | $9.2_{\pm2.7}$ | $2.2_{\pm0.4}$ | $2.2_{\pm0.4}$ | $5.4_{\pm0.9}$ | 6/6 |
| | Sym-M-CBE (2) | $3.572_{\pm0.166}$ | $24.606_{\pm1.502}$ | $9.8_{\pm2.2}$ | $5.6_{\pm0.9}$ | $22.6_{\pm7.1}$ | $4.8_{\pm0.4}$ | $3.6_{\pm0.9}$ | $9.8_{\pm2.2}$ | 6/6 |
| | Sym-M-CBE (3) | $2.025_{\pm0.253}$ | $9.277_{\pm2.933}$ | $18.6_{\pm0.9}$ | $10.8_{\pm0.4}$ | $47.8_{\pm2.7}$ | $8.8_{\pm0.4}$ | $7.8_{\pm0.4}$ | $18.6_{\pm0.9}$ | 6/6 |
| | Sym-M-CBE (4) | $1.397_{\pm0.154}$ | $4.173_{\pm0.959}$ | $24.0_{\pm0.0}$ | $14.0_{\pm0.0}$ | $60.0_{\pm0.0}$ | $12.0_{\pm0.0}$ | $10.0_{\pm0.0}$ | $24.0_{\pm0.0}$ | 0/6 |
| | Sym-M-CBE (5) | $1.528_{\pm0.366}$ | $5.107_{\pm2.640}$ | $25.2_{\pm2.5}$ | $14.2_{\pm0.5}$ | $65.8_{\pm11.5}$ | $12.0_{\pm0.0}$ | $10.5_{\pm1.0}$ | $25.2_{\pm2.5}$ | 0/6 |

*Table 13.* Concept accuracy across methods and datasets.

| Model | AWA2 (Accuracy) | AWA2-Incomplete (Accuracy) | CUB200 (Accuracy) | CUB200-Incomplete (Accuracy) | CIFAR10 (Accuracy) | dSprites-Exp. (MAE) | Pendulum (MAE) | MNIST-Arith. (MAE) | MAWPS (MAE) |
|-------|------|------|------|------|------|------|------|------|------|
| CEM | $99.54_{\pm0.01}$ | $99.22_{\pm0.02}$ | $95.70_{\pm0.38}$ | $95.50_{\pm0.24}$ | $79.29_{\pm0.09}$ | $0.1235_{\pm0.0042}$ | $0.1568_{\pm0.0193}$ | $0.6370_{\pm0.0499}$ | $0.1953_{\pm0.0039}$ |
| LICEM | $99.59_{\pm0.01}$ | $99.30_{\pm0.03}$ | $94.26_{\pm0.51}$ | $95.31_{\pm0.55}$ | $79.07_{\pm0.12}$ | $0.1264_{\pm0.0008}$ | $0.1343_{\pm0.0302}$ | $0.5789_{\pm0.0358}$ | $0.2005_{\pm0.0084}$ |
| DCR | $99.52_{\pm0.06}$ | $99.25_{\pm0.01}$ | $96.49_{\pm0.11}$ | $95.91_{\pm0.09}$ | $79.33_{\pm0.31}$ | – | – | – | – |
| CMR | $99.54_{\pm0.00}$ | $99.25_{\pm0.01}$ | $95.14_{\pm0.02}$ | $95.80_{\pm0.02}$ | $78.53_{\pm0.11}$ | – | – | – | – |
| MLP-M-CBE | $99.54_{\pm0.00}$ | $99.26_{\pm0.01}$ | $95.25_{\pm0.02}$ | $95.97_{\pm0.03}$ | $79.06_{\pm0.02}$ | $0.1230_{\pm0.0021}$ | $0.0733_{\pm0.0022}$ | $0.9157_{\pm0.0071}$ | $0.2554_{\pm0.0041}$ |
| Prior-M-CBE | – | – | – | – | – | $0.1231_{\pm0.0013}$ | $0.0652_{\pm0.0007}$ | $0.9150_{\pm0.0116}$ | $0.2523_{\pm0.0028}$ |
| Lin-M-CBE | $99.54_{\pm0.00}$ | $99.26_{\pm0.01}$ | $95.24_{\pm0.02}$ | $95.87_{\pm0.02}$ | $79.09_{\pm0.03}$ | $0.1235_{\pm0.0020}$ | $0.0753_{\pm0.0033}$ | $0.9266_{\pm0.0083}$ | $0.2558_{\pm0.0045}$ |
| Sym-M-CBE | $99.52_{\pm0.00}$ | $99.18_{\pm0.02}$ | $95.30_{\pm0.03}$ | $95.51_{\pm0.27}$ | $78.84_{\pm0.03}$ | $0.1456_{\pm0.0034}$ | $0.0728_{\pm0.0057}$ | $1.0192_{\pm0.0342}$ | $0.3528_{\pm0.0238}$ |

