# OpenReview forum: "Mixture of Concept Bottleneck Experts"
_ICML.cc/2026/Conference — ICML 2026 spotlight_

### Official Review · Reviewer_i7gZ · 2026-03-10

**Soundness:** 2
**Presentation:** 2
**Significance:** 2
**Originality:** 3
**Overall Recommendation:** 3
**Confidence:** 4

**Summary:**

This paper proposes a variant of CBM, namely Mixture of Concept Bottleneck Experts (M-CBE), and it has two variants: Linear M-CBE and Symbolic M-CBE. The authors introduce the design of these two frameworks and test these two models on a variety of datasets including image classification tasks, regression tasks, as well as math word problems. The authors use experiments to show the advantage of M-CBE in accuracy, interpretability, intervention, and adaptability compared to the baselines.

**Compliance With Llm Reviewing Policy:**

Affirmed.

**Key Questions For Authors:**

(1) Figure 2 shows a nice example for Symbolic M-CBE on a math word problem. However, its advantage is unclear on classification tasks, especially when images are complex. Is there a good example of using this variant on other tasks?

(2) Typo: line 132, "Equation **(1)** can be rewritten in conditional form as Equation (4)."

**Limitations:**

Please refer to weaknesses. The significance of the proposed models can be further explained, for instance by addressing (1) the advantage of Symbolic M-CBE and (2) the meaning/interpretation of learned expertise in M-CBE.

**Strengths And Weaknesses:**

**Strengths**:

(1) The framework provides a unified definition and can be applicable to both regression and classification tasks.

(2) The experiments and evaluation are sufficient and show promising results. (3) The design of the symbolic variant gives more intervention options to users.


**Weaknesses**:

(1) The advantage of Symbolic M-CBE is not well addressed compared to the linear variant. The design intention is well presented. However, the operations that can be used in intervention seem to be naïve, i.e., only choosing from a few operations (Section 5.3). Moreover, as shown in Table 7, the Symbolic M-CBE cannot simplify the expression, or is it adding another operator such as $^2$ ? In Table 8, using Symbolic M-CBE seems to make the expression tree deeper and more complex.

(2) Following the first weakness, the error of Symbolic M-CBE seems to be escalated if the operation is chosen carelessly. What would be a practical use case suitable for using this variant?

(3) Given there are several experts, how different are they? Are they distinguishable from each other? Would these experts (and their expertise) be interpretable? It may be worth exploring and explaining what they learned.

(4) In the evaluation of intervention (Figure 6), the CEM baseline seems to perform the best on these classification datasets.

---

> ### Author Rebuttal · Authors · 2026-03-31
>
> We thank the reviewer for their helpful feedback and for pointing out the typo.
>
> > **W1.a** *"The advantage of Symbolic M-CBE and expression complexity."*
>
> By discovering nonlinear mappings restricted to user-defined operators, Symbolic M-CBE can better align with the true underlying mechanism. This improved alignment significantly enhances interventions (Figure 6) and reduces human burden by achieving competitive accuracy with lower complexity (Figure 4).
>
> > **W1.b** *"The operations that can be used in intervention seem to be naïve [...]"*
>
> The operator vocabulary $\mathcal{W}$ is not fixed: users can freely expand it to include any higher-level functions aligned with their domain knowledge. In our experiments, we used the following vocabularies: for regression, it expands to {+, −, $\times$, sin, cos, exp, log, tan, tanh, $x^2$, $x^3$, $\sqrt{x}$, $x^{-1}$}; for classification, it is restricted to {+,-,$\times$}, since additional nonlinearities are unnecessary given binary concepts. In Section 5.3: Small = {+,-}; Medium = {+,-,$\times$}; Complete = the regression set above.
>
> We also note that, in response to reviewer **ZKVX**, we implemented a Neuro-Symbolic variant with $\mathcal{W}$ = {AND,OR,NOT}.
>
> > **W1.c** *"[...] the Symbolic M-CBE cannot simplify the expression [...]"*
>
> Symbolic M-CBE does not simplify expressions post-hoc, as optimizations such as algebraic manipulations are beyond the current scope of M-CBEs. The appearances of squared terms (e.g., $black^2$) in Table 7 were typos in the AWA2-Incomplete block. We thank the reviewer for catching this.
>
> > **W1.d** *"Symbolic M-CBE seems to make the expression tree deeper [...]"*
>
> PySR's evolutionary search jointly minimizes prediction error and node count. This yields node-compact expressions that may occasionally trade width for depth. Depth is just one complexity proxy; crucially, Symbolic M-CBE consistently appears on the Pareto front across most metrics (Tables 8–12). Furthermore, users can directly incorporate custom complexity metrics (e.g., heavily penalizing depth) into the search.
>
> > **W2.a** *"the error of Symbolic M-CBE seems to be escalated if the operation is chosen carelessly."*
>
> We acknowledge this point; like any machine learning model, Symbolic M-CBE must balance expressiveness: overly complex operator sets may lead to overfitting, while overly restrictive sets can cause underfitting.
>
> Regarding the latter, our mixture mechanism natively compensates for it. By increasing the number of experts $M$, each expert can focus on a specific input region using a simpler function to collectively approximate the mapping. This behavior is theoretically grounded (Proposition 1) and empirically verified (Figure 4). As shown in the table below, even when using a restricted "Small" operator set (see definition above), increasing \$M\$ reduces MAE:
>
> |$\mathcal{W}$|M=1|M=3|M=5|M=7|
> |-|-|-|-|-|
> |Small|3.80 ± 0.02|1.32 ± 0.02|0.87 ± 0.08|0.66 ± 0.02|
>
> We will add a dedicated appendix extending Section 5.3 with these results alongside those for the Medium and Complete sets.
>
> > **W2.b + Q1**  *"What would be a practical use case..."+ "Example of Symbolic M-CBE on complex classification."*
>
> We acknowledge that we do not have a dedicated experiment on this specific setting. Consider a dermoscopic image classification task, say melanoma vs. benign nevus, in which the concept encoder extracts clinical features (e.g., asymmetry, color variation).
>
> The two classes follow qualitatively different diagnostic mechanisms — melanoma risk is driven by the nonlinear co-occurrence of asymmetry and color variation [1], a relationship that Linear M-CBE can only approximate by increasing the number of experts $M$, ultimately raising human burden and drifting away from the true concept-to-task mechanism. Symbolic M-CBE instead can directly captures such interactions, achieving better alignment with fewer experts.
>
> > **W3** *"Diversity and interpretability of experts."*
>
> By construction, a 1-to-1 mapping exists between experts and expression trees (see **Q2** of reviewer **pJbf** for the explanation), enabling the evaluation of their diversity and interpretability. As shown in Tables 6–7, experts specialize in distinct input regions and learn unique and diverse concept-to-task relations. Each expert is as interpretable as its associated expression tree, whose interpretability is guaranteed by the restriction to user-approved functional forms.
>
> > **W4** *"Comparison with CEM baseline."*
>
> While CEM performs well on interventions, it relies on concept embeddings that lack explicit semantics.
>
> Furthermore, these embeddings facilitate information leakage. New experiments (see response to **ZKVX: W2.a + Q1**) show this leakage makes CEM highly sensitive to input perturbations, leading to severe accuracy and intervention degradation.
>
> [1] Kato et al. "Dermoscopy of melanoma and non-melanoma skin cancers." Frontiers in medicine, 2019.

---

> > ### Author Rebuttal · Reviewer_i7gZ · 2026-04-03
> >
> > I thank the authors for their reply. However, there are still some concerns regarding the advantages of Symbolic M-CBE, as its design intuition is not well motivated, and their advantages compared to linear variant (W1a,1b) can better addressed with human user experiments including operator selection and concept interpretation in practical use cases (W2.b + Q1).

---

> > > ### Author Response · Authors · 2026-04-06
> > >
> > > We thank the reviewer for their additional comment. The remaining concerns are addressed below.
> > >
> > > > *"Symbolic M-CBE design intuition is not well motivated"*
> > >
> > > We are not entirely sure what the reviewer means by *'design intuition'*, and we therefore hope the following clarification resolves any remaining ambiguity. Unlike its linear counterpart, Symbolic M-CBE allows the user to approximate the concept-to-task function using expression trees whose operators are drawn from $\mathcal{W}$, a vocabulary that can be freely chosen to match the user's preferences and domain knowledge. Building on this flexibility, the advantages of our methodology are reported below.
> > >
> > > > *"advantages of Symbolic M-CBE compared to the linear variant (W1a,1b) can be better addressed with human user experiments including operator selection and concept interpretation"*
> > >
> > > **We disagree with the need for a human user experiment**. The goal of Symbolic M-CBE is not to prove that humans prefer a set of operators over another (which would require a user study), but rather to design a concept-based model that can adapt to different user preferences, a purely architectural contribution which does not require a user study, as noted in [1]. The mathematical and quantitative advantages are proven in the experiments (Section 5):
> > >
> > > 1. **Improved Accuracy-Complexity trade-off (Figure 4):** Symbolic M-CBE is more accurate for regression datasets and on par with its linear counterpart for classification datasets, while obtaining lower complexity.
> > > 2. **Interventions (Figure 6, upper row):** Predictive accuracy under concept interventions is higher for Symbolic M-CBE than for its linear counterpart when the concept-to-task mapping is non-linear.
> > > 3. **Alignment (Table 1):** Symbolic M-CBE better approximates the underlying concept-to-task mapping, while remaining constrained to user-defined operators.
> > > 4. **Adaptability (Table 2):** Symbolic M-CBE allows the model to adapt to different user needs, contrary to its linear counterpart and the existing CBM baselines.
> > >
> > > [1] Murphy, K. P. "Probabilistic Machine Learning: Advanced Topics". MIT Press, 2023, Ch. 33, p. 1090.

---

### Official Review · Reviewer_Uxde · 2026-03-12

**Soundness:** 3
**Presentation:** 3
**Significance:** 3
**Originality:** 3
**Overall Recommendation:** 5
**Confidence:** 4

**Summary:**

The paper seeks to improve the accuracy and interpretability of concept bottleneck models by developing mixture-of-experts concept bottleneck models which have more expressive concept encoders and use a selector to select an expert model for a given input. The paper also shows that existing concept bottleneck models are instances of the proposed framework. The method is evaluated empirically through experiments examining the accuracy-interpretability tradeoff, intervenability, and adaptability of the proposed mixture-of-experts concept bottleneck models.

**Compliance With Llm Reviewing Policy:**

Affirmed.

**Final Justification:**

Thank you to the authors for the discussion! The score remains an "accept".

**Key Questions For Authors:**

**Q1. Number of trees and interpretability.** For a given prediction, if only one tree is selected (Section 3.2, lines 112-114), why do fewer expression trees (lower values of M) lead to higher interpretability (Section 3.2, lines 127-131)? Whether M is small or large, only one tree is selected for the prediction of one data point, so wouldn’t interpretability remain the same?

**Q2. How are operators in expression trees learned?** Figure 2 shows that the only learnable parameters are theta (red nodes). Thus, the operators (blue nodes) which are key for the expression tree are not learned but selected? How are they selected? What kinds of search methods / heuristics are used?

**Q3. Datasets without concept labels.** Typically concept bottleneck models are trained on a dataset that has concept labels. However, some experiments are performed on datasets (e.g., CIFAR10) that don’t have concept labels. In these setups, how are concepts identified in order to train the model? And how are the identified concepts verified?

**Q4. Complexity computation.** Complexity (on the x-axis of Figure 4) is computed as the total number of nodes in the expression tree (Section 5, lines 309-310). However, not all methods in the experiments use expression trees (for example, original concept bottleneck models), so how is complexity computed for these methods?

**Q5. Effect of intervention on error.** In Figure 6, why does intervention sometimes increase error, sometimes not affect error, and sometimes decrease error? During the intervention, predicted concepts are replaced by ground truth concepts (Section 5.2). For some prior methods (e.g., CBM, CMR, DCR in Figure 6), error increases or stays constant during the intervention. Does this suggest that these popular methods are not properly predicting concepts (even though they are developed to learn concepts) or can this be attributed to the intervention setup?


**Note.** For clarity and convenience during our discussion, I numbered each comment (i.e., S1, W1, Q1, etc.). It would be helpful to refer to these comment numbers during our discussion. Thank you!

**Limitations:**

Some limitations missing. See “Weaknesses” section above.

**Strengths And Weaknesses:**

## Strengths

**S1. Compelling research question.** The research question (how to balance accuracy and interpretability for concept bottleneck models) is conceptually interesting and useful in practice.

**S2. Sensible method.** The proposed solution (developing mixture of experts for concept bottleneck models) is reasonable and works well in practice.

**S3. Systematic experiments.** The proposed method is evaluated empirically and the experiments include multiple baselines (previous concept bottleneck methods) and diverse datasets. The results from the experiments indicate that the proposed method balances accuracy and interpretability well and enable stronger intervention on concepts.


## Weaknesses

**W1. Writing and presentation could be clearer.** While the high level idea is usually clearly presented, sometimes relevant concrete details seemed unclear. See “Questions” section below.

**W2. Missing limitations.** The limitations described are very tied to specific details of modeling steps. Some higher-level limitations include 1) concept bottleneck models typically require datasets with high-quality concept labels (ground truth concept annotations) which are relatively rare and difficult to compile and 2) concept bottleneck models tend to be hard to scale (larger or more complex tasks may involve hundreds or thousands of concepts). These limitations limit the applicability of the method.

---

> ### Author Rebuttal · Authors · 2026-03-30
>
> We appreciate the reviewer’s insightful comments that helped improve the paper.
>
> > **W1** *"Writing and presentation could be clearer"*
>
> In the revised manuscript, we will add additional details on model training, complexity computation, and concept identification to the main body. Specifically, we incorporated the explanations provided below for Q1–Q5 into our paper to improve overall clarity and reproducibility.
>
> > **W2** *"Missing limitations."*
>
> We agree with both higher-level limitations the reviewer raises. We note, however, that neither is specific to M-CBEs: both the reliance on concept annotations and scalability constraints are inherent to concept-based models in general. That said, we would like to highlight the following:
>
> - **Concept annotations**: while ground-truth labels remain the gold standard, recent label-free methods [1,2,3] demonstrate that M-CBEs can operate compatibly with automated concept extraction, as our CIFAR-10 experiment shows.
>
> - **Scaling**: unlike existing architectures (DCR, CMR), M-CBEs already scale to ~100 concepts (CUB200, CIFAR-10) without degradation.
>
> Scaling further to thousands of concepts remains an open challenge we will explicitly acknowledge in the limitations.
>
> > **Q1** *"If only one tree is selected per input, why does smaller M yield higher interpretability?"*
>
> The key distinction is between local and global interpretability:
>
> - **Local** interpretability: understanding why the model made a specific prediction. This is independent of M, since only one tree is executed per input.
> - **Global** interpretability: the ability for a human to exhaustively inspect all decision mechanisms the model could ever apply. This is directly controlled by M — with M expressions in total, a human must inspect exactly M expressions to fully understand the model's behavior.
>
> Smaller M thus reduces the human verification burden. This property is critical in safety-critical applications, where a human auditor must certify that no possible reasoning path is incorrect, not just the one used for a given sample. We will clarify this distinction in the revised manuscript.
>
> > **Q2** *"How are operators in expression trees learned?"*
>
> To learn the interpretable expression trees (operators included), we first instantiate a placeholder MLP for each expression tree. This allows the routing mechanism $p(t\mid x)$ to partition the input space into regions, each assigned to a different MLP. We then apply symbolic regression — specifically, the multi-population evolutionary algorithm in PySR, which jointly minimizes prediction error and expression complexity — to learn the concept-to-task mapping for the input region specified by $p(t\mid x)$.
>
> An end-to-end gradient-based alternative using KANs is provided in Appendix D; however, we found PySR more tractable and that it yielded lower-complexity expressions (Table 4).
>
> > **Q3** *"Datasets without concept labels."*
>
> For CIFAR-10, which lacks concept annotations, we apply the label-free method of Oikarinen et al. [1] to automatically extract binary concept labels. This experiment is specifically designed to demonstrate M-CBE's compatibility with label-free pipelines.
>
> > **Q4** *"Complexity computation."*
>
> Methods that can be considered instances of M-CBEs (including CBM, CMR, DCR, LICEM) are formalized as specific expression trees in our framework, so node count is well-defined for all of them (Appendix C). For example, CBM is a Linear M-CBE with M=1, whose linear tree complexity is counted directly. For methods whose task predictor does not operate on concept predictions in an interpretable form, specifically CEM, which uses concept embeddings, and BlackBox,  expression tree complexity is undefined. This is why they appear as horizontal lines across the full x-axis in Figure 4.
>
> > **Q5** *"Effect of intervention on error."*
>
> The varying impact of interventions directly highlights whether a model has truly captured the correct underlying data-generating mechanism. When an intervention increases or does not affect the error, it indicates that the task predictor has failed to align with the true concept-to-task logic and may instead rely on information leaked from the input.
>
> On MNIST-Arithm, a single linear layer cannot represent multiplication or division. In such cases, the model usually compensates for this structural limitation by finding shortcuts through information leaked from the input. Because the predictor relies on this leakage rather than the concepts, even perfect interventions fail to reduce error. We provide a detailed experiment on this in our response to **ZKVX (Q1)**.
>
> [1] Oikarinen et al. "Label-free concept bottleneck models". ICLR, 2023.
> [2] Yang et al. "Language in a bottle.". CVPR, 2023.
> [3] Srivastava et al. "VLG-CBM.". NeurIPS, 2024.

---

> > ### Author Rebuttal · Reviewer_Uxde · 2026-04-01
> >
> > Thank you to the authors for addressing points raised in the review! The score assigned remains an "accept".

---

> > > ### Author Response · Authors · 2026-04-05
> > >
> > > We thank the reviewer for acknowledging our rebuttal and confirming their positive assessment.

---

### Official Review · Reviewer_ZKVX · 2026-03-13

**Soundness:** 3
**Presentation:** 3
**Significance:** 2
**Originality:** 2
**Overall Recommendation:** 4
**Confidence:** 4

**Summary:**

The paper aims at improving concept-based methods by enriching the
task prediction module, introducing multiple experts and intepretable
classes of functions (linear functions and symbolic regression).

**Compliance With Llm Reviewing Policy:**

Affirmed.

**Final Justification:**

This work contributes to the panorama of concept-based models by boosting their predictive performance thanks to mixture of expert + non-trivial interpretable classification layers. While this implies a trade-off between interpretability and accuracy that deserves further investigation, the  experimental evaluation, further strengthened by the experiments ran during the rebuttal, confirms the advantange of the proposed solution over existing alternatives.

**Key Questions For Authors:**

- You justify mixture of experts to allow context-dependent reasoning. Isn't this hinting at instance-level explanations? What is the advantage of this intermediate solution?

- How does the approach relate to neuro-symbolic solutions (which have the advantage of a principled framework for reasoning on top of concepts)? can't symbolic regression (or any other rule learning approach) be used to learn the reasoning layer there instead?

**Limitations:**

yes

**Strengths And Weaknesses:**

PROs

Concept-based models suffer from limited applicability because of
sub-optimal performance and need for concept-level supervision
(partially mitigated by CLIP-generated concepts). This work
contributes to addressing the former problem.

The proposed solution is backed by theoretical results on expressivity
and approximation guarantees.

The experimental evaluation is extensive and carefully designed.

CONs

The originality of the proposed solution is not dramatic, as it
consists in replacing the linear layer of CBMs with a mixture of
well-known interpretable functions (linear functions or symbolic
regression trees).

The practical utility of the framework is not entirely clear. The LICEM approach
already achieves minimal error. It is regarded as having infinite
(global) complexity because it produces a distinct explanation for
each example. However it is unclear whether this instance-level
perspective (which is shared by most post-hoc explainers) on
explanations is considered less interpretable by humans. Additionally,
the introduction of the mixture component further complicates interpretability.
These aspects should be better discussed.


Minor:
In figure 4, the line type for baselines differs between plot and legend, which is confusing

---

> ### Author Rebuttal · Authors · 2026-03-30
>
> We thank the reviewer for their helpful comments and feedback.
>
> > **W1** *"The originality of the proposed solution is not dramatic"*
>
> We respectfully disagree. In our view, the novelty lies at three complementary levels:
>
> 1. **Theoretical Unification**: Our framework formalizes and unifies a fragmented landscape of CBM architectures (Figure 1).
>
> 2. **Novel, State-of-the-Art Instantiations**: We introduce two concrete models, Lin-M-CBE and Sym-M-CBE, that outperform existing globally interpretable baselines in predictive accuracy across multiple benchmarks.
>
> 3. **Systematic Design Space Exploration**: By decoupling functional form from the number of experts, we conduct a systematic exploration of a previously uncharted 2D design space. Notable findings include:
>
>     - Boolean forms scale poorly with bottleneck size.
>     - Multiple experts compensate for incomplete bottlenecks.
>     - Over-parameterized experts (e.g., MLPs) are Pareto-dominated.
>
> Our work establishes a sound foundation for designing concept-based architectures, with utility demonstrated by the two novel instantiations, while offering practical guidance for navigating the accuracy-interpretability trade-off.
>
> > **W2.a + Q1**  *"LICEM already achieves minimal error." + "What is the advantage of this intermediate solution?"*
>
> While M-CBEs provide instance-level explanations, they also preserve global interpretability. This property is missing in methods like LICEM, which leads to important limitations:
>
> 1. **Reasoning shortcuts.** Since each sample admits many valid explanations, these models are not identifiable. This means they can discover equations that are mathematically valid but semantically meaningless [1].
>
> 2. **Verifiability.** LICEM's decision mechanism cannot be inspected without a specific sample. In contrast, M-CBEs' finite set of expressions allows a human to verify all reasoning paths before deployment.
>
> 3. **Knowledge Integration.** LICEM produces a different expression for each input, so there is no fixed formula that users can constrain beforehand. M-CBEs address this directly, as demonstrated by Prior-M-CBE, which can integrate user-defined expressions.
>
> 4. **Robustness.** LICEM's continuous parameterization allows residual input information to bypass concepts, degrading accuracy and intervention responsiveness even under small perturbations [2]. The table below confirms this: under Gaussian input noise, LICEM's MAE degrades sharply and fails to recover even when all concepts are corrected ($p_{int}=1.0$), while Sym-M-CBE fully recovers ($\text{MAE}=0.00$).
>
>
> |$p_{\text{int}}$|**MAWPS**|  | | **MAWPS (perturbed input)** | |  |
> | - | - | - | - | - | - | - |
> | | CEM | LICEM | Sym-M-CBE    | CEM | LICEM | Sym-M-CBE  |
> | 0.0 | 0.73 ± 0.02  | 0.74 ± 0.03  | 1.33 ± 0.18 | 14.79 ± 0.51 | 30.02 ± 7.69 | 8.03 ± 1.97  |
> | 0.5| 0.43 ± 0.01  | 0.52 ± 0.02  | 0.66 ± 0.09 | 10.16 ± 0.44| 17.98 ± 4.17 | 4.06 ± 1.00  |
> | 1.0| 0.13 ± ≤0.01 | 0.31 ± ≤0.01 | 0.00 ± ≤0.01 | 5.53 ± 0.47| 6.28 ± 0.64   | 0.00 ± ≤0.01 |
>
> We thank the reviewer for raising this point. We think this experiment will strengthen the paper and will report it in a dedicated appendix.
>
> > **W2.b** *"the introduction of the mixture component further complicates interpretability."*
>
> There might be a misunderstanding. The reasoning process remains transparent: users see the **finite set** of interpretable expert expressions and inspect which one is selected for each prediction.
>
> > **Q2** *"How does the approach relate to neuro-symbolic solutions [...] can't symbolic regression [...] be used to learn the reasoning layer there instead?"*
>
> M-CBEs are themselves a neuro-symbolic approach, extending standard methods along two dimensions: *functional form*, generalizing beyond Boolean logic to user-defined operators, and *expert cardinality*, using  multiple symbolic experts instead of a single global rule.
>
> This flexibility naturally permit to include rule-based reasoning: setting the operator vocabulary to $\mathcal{W}$ = {AND, NOT, OR} instantiates Rule-M-CBE, which outperforms both DCR and CMR in accuracy and accuracy under interventions ($p_{int}>0$) on CUB200 (table below).
>
> | $p_{\text{int}}$ | DCR | CMR | Rule-M-CBE   |
> | - | - | - | - |
> | 0.0| 19.77 ± 2.46 | 0.49 ± 0.28 | 41.14 ± 0.08  |
> | 0.5| 22.34 ± 3.07 | 0.51 ± 0.28 | 70.74 ± 0.42  |
> | 1.0 | 24.70 ± 3.74 | 0.52 ± 0.32 | 100.00 ± ≤ 0.01 |
>
> We speculate this gap is due to two factors: first, DCR and CMR apply heavy regularization to preserve interpretability; second, they restrict the learned rules to conjunctions of positive or negated concepts, a strictly more limited functional form w.r.t. Rule-M-CBE.
>
> We thank the reviewer for this question. The Rule-M-CBE variant will be added to appendix H.
>
> [1] Debot, David, et al. "Interpretable concept-based memory reasoning". NeurIPS, 2024.
> [2] Espinosa Z. et al. "Avoiding leakage poisoning". ICML, 2025.

---

> > ### Author Rebuttal · Reviewer_ZKVX · 2026-04-03
> >
> > I would like to thank the authors for the detailed answer and the additional experiments, which help clarifying the practical advantage of the proposed solution over both instance-level and model-level alternatives. I am still undecided about the advantage of having a mixture-of-expert solution (which can imply a finite but rather large set of explanations to look at for deployment, what if only some are acceptable?) and an instance-level one that allows instance-level decision about acceptability, but I think this discussion goes beyond the scope and merits of this paper, and requires human evaluations to achieve any sensible conclusion. I will raise my score.

---

> > > ### Author Response · Authors · 2026-04-06
> > >
> > > We thank the reviewer for the positive feedback and constructive discussion. We will make sure to properly integrate the discussed analysis in the revised paper.
> > >
> > > Regarding which actions can be taken if some expressions are not acceptable, we think that this paper paves the way for several possibilities. For example, the problematic expressions can be dropped from the set and the selector retrained to select only over the remaining expressions. We leave as future work an exhaustive analysis of the possible action items.

---

### Official Review · Reviewer_pJbf · 2026-03-13

**Soundness:** 3
**Presentation:** 3
**Significance:** 3
**Originality:** 3
**Overall Recommendation:** 6
**Confidence:** 4

**Summary:**

Authors propose a family of novel interpretable machine learning models, building up on the field of concept-based explainable AI, and the popular mixture-of-experts mechanism. Concretely, they introduce expression trees for concept bottleneck models, unlocking symbolic capabilities and control from the user, as well as adding multiple models together. The novel architecture remains a universal approximator, and is interpretable both through the nature of the parameters (i.e. concepts) and through the nature of the operators manipulated. Furthermore, they remain actionable and intervenable (with background knowledge or human corrections). Experiments show that M-CBEs are pareto-optimal for the complexity-accuracy trade-off, and are competitive.

**Compliance With Llm Reviewing Policy:**

Affirmed.

**Final Justification:**

The weaknesses and questions I had for the authors were fully addressed during rebuttal, and I am convinced their answers (plus the experiments they had in supplementary) to other reviewers will improve the quality of the paper

**Key Questions For Authors:**

1. While reasonably out-of-scope for this paper, did you see if this method improves the quality of concept identification $p(c|x)$ ?
2. If I understand well, each expert $m$ has different expression trees (or influences it) through $p(t|m)$ ?
3. I am skeptical about the true interpretability of the (variants of) M-CBEs:
3.a. You add MoE to CBMs, which naturally improves the accuracy of those models. However, how interpretable does it become when you have to "explain" why a certain tree/linear model/MLP is chosen $p(t | x)$  ?
3.b. Similarly, how interpretable is it to say why an expert was chosen $p(m|x)$ ?
3.c. Finally, how interpretable is it, depending of course on which operators were chosen from $\mathcal W$, when the function becomes $\alpha\cdot\sin(c_{cat-ears}) \cdot \exp(c_{cat-eyes})$.
I am not convinced _right now_ that such nonlinear transformations respect the inherent _trust_ we have in CBMs / CBEs that such "neuron" is truly representative of the concept "cat ears" or "parrot beak".

**Limitations:**

yes

**Strengths And Weaknesses:**

# Strengths

1. The two main ideas are clearly laid out and explained, the submission is overall well-written and clear.
2. The propositions supported by clear proofs.
3. Figures drive the understanding of the reader.
4. Strong empirical evidence is provided to support the claims.
5. Authors show how certain properties (like intervenability or adaptability) can be measured against previous CBMs, and 'oracle' version of the methods are evaluated along.

# Weaknesses

1. A great amount of work is spent to motivate and prove how Symbolic M-CBEs can work and are better **but** it is not clear which operators from $\mathcal W$ are selected in the experimental section, nor is it properly explained how to train them (page 5 roughly describes textually how it is done, but not in depth)
2. The interpretability of the model is expressed only through its complexity, no attempt to measure it with different ways.

---

> ### Author Rebuttal · Authors · 2026-03-30
>
> We thank the reviewer for their constructive and positive feedback.
>
> >  **W1** *"[...] it is not clear which operators are selected in the experimental section, nor is it properly explained how to train them."*
>
> We appreciate this feedback and will strengthen the main text by incorporating the details about operator selection and training procedures from Appendix E and F.
>
> Specifically, the training follows a three-stage pipeline. First, we jointly train the concept encoder, selector, and placeholder MLP experts end-to-end by maximizing Eq. (5), enabling the selector to partition the input space while learning concepts and task mappings. Second, we replace each expert with a symbolic expression obtained via symbolic regression on its assigned data. We use PySR (40 populations, size 60, 100 iterations), restricting operators to {+, −, $\times$} for classification and using a richer set {+, −, $\times$, sin, cos, exp, log, tan, tanh, $x^2$, $x^3$, $\sqrt{x}$, $x^{-1}$} for regression. Finally, we fine-tune the model end-to-end, keeping the symbolic structure fixed and updating only numerical parameters together with the concept encoder and selector.
>
> > **W2** *"The interpretability of the model is expressed only through its complexity"*
>
> We agree that the current wording in the "Metrics" section may give the impression that interpretability is assessed solely through complexity. In practice, however, we evaluate interpretability through three complementary lenses:
>
> 1. **Model Complexity**: Node count, depth, and variable/operator count, following the principle that brevity aids comprehension [1].
>
> 2. **Concept Interventions**: Interventions reveal how much and how well the task predictor leverages concepts [2], providing direct insight into the model's decision-making mechanism. As shown in Figure 6 in the manuscript, both M-CBE variants exhibit strong responsiveness to concept interventions, indicating that the discovered expressions use concepts as intended.
>
> 3. **User Adaptability**: A compact expression tree is not sufficient for interpretability if it contains symbols that are unintelligible to the user. Therefore, user adaptability is a necessary (though not sufficient) condition for interpretability.
>
> To address this feedback, we will update Sec. 5 to clarify our evaluation.
>
> > **Q1** *"[...] this method improves the quality of concept identification $p(c\mid x)$?"*
>
> This does not appear to be the case: Table 13 (Appendix K.2) shows that the accuracy of concept predictions is similar across all methods, with no clear advantage for any approach.
>
> > **Q2** *"If I understand well, each expert $m$ has different expression trees (or influences it) through $p(t\mid m)$."*
>
> It is correct that experts influence the symbolic expressions; however, in our current formulation, **each expert $m$ is mapped to a single expression tree $t_m$**, rather than multiple trees.
>
> Formally, $p(t \mid m)$ is modeled as a degenerate distribution that assigns all probability to a single tree $t_m$. We made this design choice to ensure that model expressivity is governed solely by two factors: the user constraints over the expression tree (e.g., operator vocabulary $\mathcal{W}$) and the number of experts $M$. Allowing multiple trees per expert would introduce an additional hyperparameter — the number of trees per expert — that would also govern expressivity.
>
> >**Q3.a +Q3.b** Interpretability of the selection process
>
> We agree with the reviewer that the expression tree selection process is opaque, as noted in the limitations. However, competing methods such as LICEM and DCR also perform input-dependent selection implicitly (Appendix C), without producing a finite and inspectable expression set, making M-CBEs comparatively more transparent.
>
> We will update Sec. 3.2 to briefly explain this implicit selection.
>
> > **Q3.c** *"[...] I am not convinced right now that such nonlinear transformations respect the inherent trust we have in CBMs / CBEs [...]"*
>
> To maintain interpretability, **users can eliminate operators deemed inappropriate for specific concepts**. In our discrete setting, we restrict $\mathcal{W}$ = {+, -, $\times$}. While nonlinear interactions arise via multiplication, these represent intuitive concept conjunctions (e.g., $c_1 \times c_2$) rather than opaque transformations, preserving the inherent trust in the model.
>
> For clarification, we note that concept semantics are independent of the expression tree. The meaning of $c_i$ is determined solely by the concept encoder $p(c\mid x)$, trained to predict ground-truth concept values (concept accuracy shown in Table 13). The expression tree operates downstream of this bottleneck and cannot alter what $c_i$ represents. Therefore, the semantics of the concepts in M-CBE are the same as in "traditional" CBMs.
>
> [1] Miller, G. A. "The magical number seven". Psychological review, 1956.
> [2] Koh, Pang Wei, et al. "Concept bottleneck models". ICML, 2020.

---

> > ### Author Rebuttal · Reviewer_pJbf · 2026-04-04
> >
> > The authors have answered my questions and I am satisfied. I encourage them to highlight more their methodology for evaluating CBMs / CBEs (W2) ; explain more that each expert has only one tree associated with it (Q2) ; and discuss a bit, as they have done with me, the interpretability of the operators and expression trees (Q3).
> >
> > I will raise my score.

---

> > > ### Author Response · Authors · 2026-04-05
> > >
> > > We thank the reviewer for the positive assessment and the constructive feedback. We are happy to integrate all three suggestions into the revised manuscript.

---

### Decision · Program_Chairs · 2026-04-30

**Decision:**

Accept (spotlight)

**Comment:**

This paper proposes a mixture of concept bottleneck experts, a method for interpretable supervised models in both regression and classification that attempts to alleviate the limitation of traditional CBMs in deteriorating performance at the expense of interpretability. The authors use a linear and a symbolic mixture of CBEs, provide an analysis of the tradeoff of accuracy and interpretability, intervention behavior and adaptability across different tasks.

The is broad agreement among reviewers that this paper addresses and important problem of tradeoffs between interpretability and accuracy in CBMs. The paper has a balanced presentation of theory and empirical validation, which was quite thorough and comprehensive. Some concerns were also raised, chief among being that the introduction of a mixture of experts introduces less interpretability, and there were common questions about the specifics of the symbolic M-CBE operator. The rebuttal process was overall productive, and most of the reviewers found their concerns either completely resolved, or partially resolved. One reviewer did not find the authors' responses fully satisfying, and remains unsupportive.

After reviewing the comments, responses, and the paper, I agree that the degree of responses from the authors have alleviated most of the key concerns. I think this paper provides a nice balanced presentation on an elegant method that generalizes CBMs.